# Parameters as Experts: Adapting Vision Models with Dynamic Parameter Routing

Meng Lou [1]  Stanley T. Yu [2]  Yizhou Yu [1]

## Abstract

Adapting pre-trained vision models using parameter-efficient fine-tuning (PEFT) remains challenging, as it aims to achieve performance comparable to full fine-tuning using a minimal number of trainable parameters. When applied to complex dense prediction tasks, existing methods exhibit limitations, including input-agnostic modeling and redundant cross-layer representations. To this end, we propose ParaX, a new adapter-style method featuring a simple mixture-of-experts (MoE) architecture. Specifically, we introduce shared expert centers, where each expert is a trainable parameter matrix. During a feedforward pass, each ParaX module in the network dynamically generates weight matrices tailored for the current module via a simple dynamic parameter routing mechanism, which selectively aggregates parameter matrices in the corresponding expert center. Dynamic weight matrices in ParaX modules facilitate low-rank adaptation in an input-dependent manner, thus generating more customized and powerful feature representations. Moreover, since ParaX modules across multiple network layers share the same expert center, they improve feature diversity by promoting implicit cross-layer feature interaction. Extensive experimental results demonstrate the superiority of ParaX across diverse visual recognition tasks. Code is publicly released at: https://github.com/LMMMEng/ParaX.

## 1. Introduction

Parameter-efficient fine-tuning (PEFT) (Han et al., 2024) aims to update or embed only a small number of parameters into a pre-trained model while performing comparably to

full fine-tuning. This approach has been widely adopted in both natural language processing (NLP) and computer vision. For instance, prompt-based tuning in NLP tasks (Liu et al., 2023) has inspired many PEFT methods in vision. A representative work is VPT (Jia et al., 2022), which inserts a set of learnable tokens into the input sequence of Vision Transformers (ViTs) (Dosovitskiy et al., 2021; Liu et al., 2021), achieving task adaptation with minimal additional parameters. Although prompt-based tuning methods demonstrate promising performance on classification tasks, there still exists a considerable performance gap between these methods and full fine-tuning on more complex vision tasks, such as dense predictions. On the other hand, adapter-based PEFT methods (Houlsby et al., 2019) have also attracted considerable attention. A well-known example is LoRA (Hu et al., 2022), which learns low-rank adapters to achieve efficient fine-tuning of large language models (LLMs). In the same spirit, AdaptFormer (Chen et al., 2022) introduces a lightweight MLP module to adapt ViTs, representing an early attempt to utilize adapters in visual recognition. Subsequently, Mona (Yin et al., 2025) further integrates multi-scale depthwise convolutions into the adapter module to enhance its spatial modeling capacity for dense predictions. Although existing adapter-based methods have achieved promising results in diverse vision tasks, two key challenges remain unresolved:

**Representation Deficiency**. As shown in Figure 1 (a), each adapter is responsible for task-specific model adaptations using input-agnostic low-rank modeling. For complex tasks such as dense predictions, it is inherently challenging to learn task-specific transformations that work universally well for all possible inputs. The fact that such adapters cannot support input-dependent adaptations to account for input variations limits the feature representation capacity of the resulting model. To empirically verify this, we use the effective receptive field (ERF) (Luo et al., 2016) to visualize a model's representation capacity. Specifically, we fine-tune the Swin-L model (Liu et al., 2021) pre-trained on ImageNet-21K (Deng et al., 2009) on the COCO2017 dataset (Lin et al., 2014) using the Mask R-CNN framework (He et al., 2017). As shown in the first row of Figure 1 (c), previous representative PEFT methods, including AdaptFormer and Mona, exhibit smaller ERFs compared to full fine-tuning.

[1]The University of Hong Kong [2]University of Pennsylvania. Correspondence to: Yizhou Yu <yizhouy@acm.org>.

*Proceedings of the 43rd International Conference on Machine Learning*, Seoul, South Korea. PMLR 306, 2026. Copyright 2026 by the author(s).

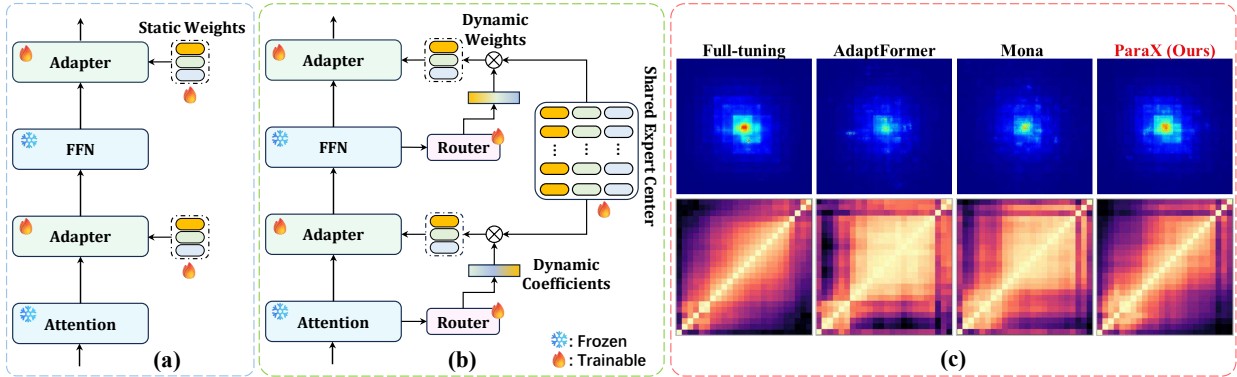

*Figure 1.* **(a)** Classical adapter-based PEFT methods (e.g., Mona (Yin et al., 2025)). **(b)** Our proposed ParaX. Normalization layers and residual connections are omitted for simplicity. **(c)** The first and second rows show ERF and CKA visualizations for various fine-tuned models, respectively. Specifically, Swin-L model pre-trained on ImageNet-21K is used as the backbone network, which is fine-tuned on the COCO2017 using various fine-tuning methods and the Mask R-CNN framework. Quantitative results are listed in Table 2.

This phenomenon indicates that these methods weaken the model's ability to capture complex spatial dependencies required for dense prediction tasks.

**Feature Redundancy**. Adapters embedded in different network layers and their parameters are isolated from each other, and such a lack of cross-layer interaction may lead to redundant representations. To illustrate this limitation, we perform a centered kernel alignment (CKA) analysis (Kornblith et al., 2019) for different methods. As shown in the second row of Figure 1 (c), the patterns learned by different layers in both AdaptFormer and Mona exhibit a higher similarity compared to those under full fine-tuning, which means that different layers capture redundant information.

To address these limitations, we propose a new adapter-based PEFT method dubbed ParaX. As illustrated in Figure 1 (b), ParaX is built upon a simple mixture-of-experts (MoE) architecture. Specifically, we construct a large shared expert center comprising a collection of trainable parameter matrices, each having the same size as the corresponding weight matrix in a standard adapter. Each ParaX module in the network dynamically generates weight matrices tailored for the current module via a dynamic parameter routing mechanism, which selectively aggregates parameter matrices in this shared expert center. This routing mechanism is analogous to the gating mechanism in MoE that selects appropriate experts for a given input, and the trainable parameter matrices are treated as experts in this work. Although our design is simple, it offers two advantages that are absent in previous works:

First, dynamic weight matrices in ParaX modules facilitate low-rank adaptation in an input-dependent manner, thus generating more customized and powerful feature representations. As evidenced in Figure 1 (c), the ERF of our model is larger than those of other PEFT methods and comparable to that of full fine-tuning. Such a large ERF enables our model to capture long-range dependencies more easily,

which is crucial in dense predictions (Xie et al., 2021; Fu et al., 2025).

Second, since the same expert center is shared among ParaX modules in multiple network layers, an implicit cross-layer feature interaction can be developed to diversify the information flow, thus reducing feature redundancy (Huang et al., 2017; Kim et al., 2024; Lou et al., 2025a). As evidenced in Figure 1 (c), the feature diversity of our fine-tuned model is better than that of other PEFT methods and very close to that of full fine-tuning. This means that our method can extract more representative features from complex scenes for dense predictions.

We have evaluated our method on a wide range of visual recognition tasks, including semantic segmentation, object detection and instance segmentation, panoptic segmentation, and image classification. Extensive experiments in Section 4 demonstrate that our method has achieved superior performance compared to previous PEFT methods. For instance, in the semantic segmentation task on ADE20K with Swin-L, ParaX surpasses full fine-tuning by 0.8% in mIoU while requiring less than 4% of trainable parameters. In the object detection and instance segmentation task on COCO2017 with ConvNeXt-L, our method achieves improvements of 0.6%/0.4% in $AP^b$/$AP^m$ over Mona, using a comparable number of parameters, and outperforms full fine-tuning by 1.4%/1.6% in $AP^b$/$AP^m$. Furthermore, we employ PEFT methods for panoptic segmentation, a more challenging task that unifies semantic segmentation, object detection, and instance segmentation, and has been underexplored in prior work. Specifically, when integrated with Swin-B and ConvNeXt-B, our method improves over AdaptFormer by 1.7% and 2.0% in PQ, respectively.

## 2. Related Work

**Efficient Transfer Learning in Language Models**. With the development of LLMs, PEFT techniques have been

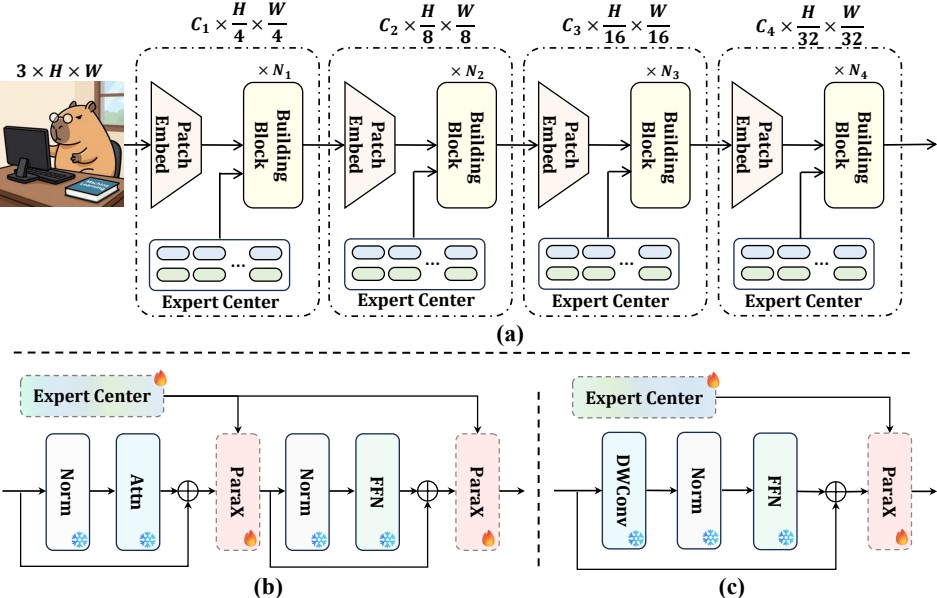

*Figure 2.* An overview of our proposed ParaX. (a) denotes a hierarchical vision model equipped with shared expert centers and ParaX. (b) and (c) refer to the Swin- and ConvNeXt-style building blocks with ParaX, respectively.

largely pioneered within the NLP community. For instance, BitFit (Ben-Zaken et al., 2022) only updates bias terms in a backbone network and parameters outside the backbone. Prompt-based tuning methods (Lester et al., 2021; Li & Liang, 2021; Liu et al., 2022a) aim to achieve task adaptation by prepending a small number of learnable tokens to the input sequence, while keeping the pre-trained weights frozen. Adapter-based methods (Houlsby et al., 2019; Pfeiffer et al., 2020; 2021; Hu et al., 2022; Wang et al., 2022; Liu et al., 2024a) embed small trainable modules within the layers of a frozen pre-trained network. A highly influential approach is LoRA (Hu et al., 2022), which approximates weight updates via low-rank matrices. Subsequently, SMoP (Choi et al., 2023) sparsely activates a set of prompts using a gating mechanism. MoELoRA (Luo et al., 2024) employs contrastive learning to encourage distinct feature learning among experts, mitigating random routing issues. HydraLoRA (Tian et al., 2024) decomposes a projection matrix into multiple mini-rank matrices and uses a router to combine their outputs. DoRA (Liu et al., 2024a) decomposes pre-trained weights into magnitude and direction to enhance both learning capacity and stability. MoLA (Gao et al., 2025) allocates different numbers of LoRA experts to different layers. HiRA (Huang et al., 2025) devises a Hadamard product-based LoRA to facilitate high-rank adaptation.

**Parameter-efficient Visual Recognition**. The aforementioned PEFT methods in NLP have served as a primary source of inspiration for PEFT methods in computer vision (Jie & Deng, 2022; Luo et al., 2023; Jie & Deng, 2023; Wang et al., 2024; Yin et al., 2025; Ran et al., 2025). For example, VPT (Jia et al., 2022) has successfully adapted

prompt-based tuning by prepending learnable tokens to the input sequence of ViTs. DA-VPT (Ren et al., 2025) further improves visual prompt learning through semantic metric construction between prompts and image features. TCPA (Liu et al., 2025b) assigns coordinated prompts to different tokens to facilitate attention interactions. On the other hand, adapter-based methods have also been extensively explored. AdaptFormer (Chen et al., 2022) parallels a lightweight MLP module to the original channel mixer in ViTs. KAdaptation (He et al., 2023b) decomposes and updates adapter weights through the Kronecker product. SPT (He et al., 2023a) adaptively selects the most sensitive weights for a given task. GPS (Zhang et al., 2024) proposes a gradient-based parameter selection strategy. MLAE (Wang et al., 2024) selectively activates low-rank matrices during training. LoRand (Yin et al., 2023) sparsely combines low-rank weights in adapters for dense predictions. SNELL (Shen et al., 2024) sparsifies trainable matrices to reduce memory costs. CoLoRA (Ran et al., 2025) designs a PEFT method tailored to ConvNets with correlated low-rank adaptation. Mona (Yin et al., 2025) introduces multi-scale spatial modeling capability into adapters by integrating multi-kernel convolutions. These methods demonstrate promising potential for efficiently adapting vision models (Yan et al., 2015; Liu et al., 2021; Woo et al., 2023; Lou et al., 2025b;a; Shi et al., 2026) to different vision tasks.

Unlike the aforementioned methods, this paper develops a large shared expert center that can be dynamically queried by layer-specific routers embedded in different network layers. This enables the construction of adapters based on a rich parameter space to facilitate cross-layer interactions, resulting in superior performance over previous PEFT methods

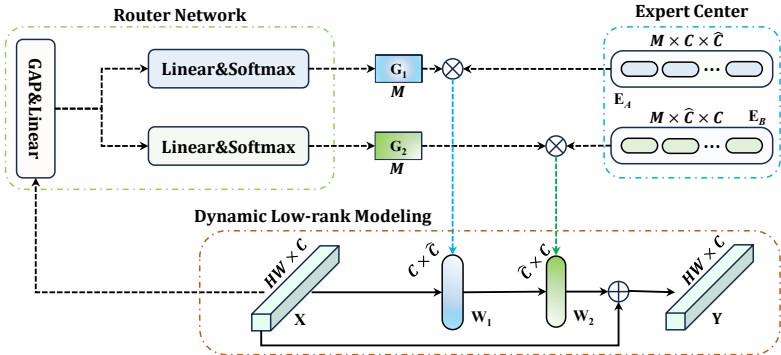

*Figure 3.* The schematic workflow of dynamic parameter routing in ParaX.

on complex vision tasks, particularly in dense predictions.

# 3. Method

## 3.1. Overview

As illustrated in Figure 2 (a), we take a typical four-stage hierarchical vision architecture as an example, where each stage consists of a patch embedding layer followed by a few network building blocks. Within each stage, we set up a large shared expert center containing a collection of trainable parameter matrices, while ParaX is integrated into every building block. For a Swin-like transformer network (Liu et al., 2021), a ParaX module is attached after every token mixer as well as every channel mixer, as shown in Figure 2 (b). In a ConvNeXt-style network (Liu et al., 2022b), since each block encapsulates the token mixer and channel mixer into a single residual module, a ParaX module is attached after every complete ConvNeXt block, as shown in Figure 2 (c). During a forward pass, each ParaX module selectively combines parameter matrices from the shared expert center in the corresponding stage via a dynamic parameter routing mechanism, analogous to the simple expert selection strategy in MoE. This process generates dynamic weight matrices, enabling input-dependent low-rank transformation of input features.

## 3.2. ParaX

**Shared Expert Center**. Each shared expert center contains a collection of trainable parameter matrices. For channel-wise transformations, the parameter matrices form pairs,

$$\left\{ \mathbf{E}_A \in \mathbb{R}^{M \times C \times \hat{C}}, \quad \mathbf{E}_B \in \mathbb{R}^{M \times \hat{C} \times C} \right\},$$

where $M$ denotes the capacity of the expert center. Both $M$ and $\hat{C}$ are hyperparameters that control the number of trainable parameters, and their configurations are discussed in Section 4.5. Since multi-scale spatial mixing is essential for adapting models in dense prediction tasks (Yin et al., 2025), for generating dynamic weight matrices for spatial transfor-

mations, we introduce another set of parameter matrices for implementing multi-kernel depthwise convolutions,

$$\{\mathbf{S}_A \in \mathbb{R}^{M \times \hat{C} \times K_1^2}, \mathbf{S}_B \in \mathbb{R}^{M \times \hat{C} \times K_2^2}, \mathbf{S}_C \in \mathbb{R}^{M \times \hat{C} \times K_3^2}\},$$

where $K_i^2$ represents one of the three kernel sizes. We discuss the combined settings of multiple kernel sizes in Section 4.5.

**Dynamic Parameter Routing**. For simplicity, let us consider the case involving channel projection only. As illustrated in Figure 3, given an input feature $\mathbf{X} \in \mathbb{R}^{HW \times C}$ (where $C$ and $HW$ denote the channel and spatial dimensions, respectively), a lightweight router network generates dynamic coefficients for the parameter matrices in the expert center. Specifically, the input feature first undergoes global average pooling (GAP), followed by a linear layer that produces a hidden representation with a significantly reduced channel dimension (i.e., 16 channels in our implementation) to minimize computational overhead. This hidden feature is then passed through two parallel linear layers with softmax activation, yielding two dynamic gating vectors $\{\mathbf{G}_1, \mathbf{G}_2\} \in \mathbb{R}^M$. These gating vectors are used to dynamically aggregate the parameter matrices in the expert center: $\mathbf{G}_1$ is multiplied with $\mathbf{E}_A$ to produce the dynamic down-projection weight matrix $\mathbf{W}_1 \in \mathbb{R}^{C \times \hat{C}}$, and similarly, $\mathbf{G}_2$ is multiplied with $\mathbf{E}_B$ to form the dynamic up-projection weight matrix $\mathbf{W}_2 \in \mathbb{R}^{\hat{C} \times C}$. Although more advanced MoE routing mechanisms exist, this simple yet efficient design aligns well with the efficiency consideration of PEFT. The dynamically composed weight matrices $\mathbf{W}_1$ and $\mathbf{W}_2$ are then used to transform the input feature $\mathbf{X}$ in an input-dependent and channel-wise manner. The final output $\mathbf{Y}$ is obtained by adding a residual connection (He et al., 2016) to this dynamically transformed input feature.

**Dynamic Multi-scale Spatial Mixing**. Inspired by Mona (Yin et al., 2025), we equip ParaX with multi-kernel depthwise convolutions to enhance the spatial mixing of latent features produced by the dynamic down-projection weight matrix $\mathbf{W}_1$. Nonetheless, the convolution kernels

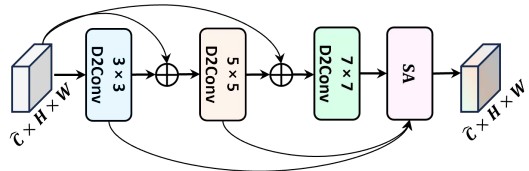

*Figure 4.* A schematic diagram of dynamic multi-scale spatial mixing.

in ParaX are dynamically generated from the shared expert center. Specifically, the router network produces three dynamic gating vectors $\{\mathbf{G}_A, \mathbf{G}_B, \mathbf{G}_C\} \in \mathbb{R}^M$, which are multiplied with the corresponding parameter matrices $\{\mathbf{S}_A, \mathbf{S}_B, \mathbf{S}_C\}$ to produce three dynamic convolution kernels. Then, these kernels are applied to the latent features via depthwise convolutions. Although the kernel generation process is conceptually similar to classical dynamic convolutions (He et al., 2019; Li et al., 2022; Li & Yao, 2024), a key difference is that our method produces dynamic depthwise convolution kernels (D2Convs) rather than standard convolution kernels.

On the other hand, our design employs a sequentially stacked multi-scale convolution structure that progressively expands the receptive field, as depicted in Figure 4. Specifically, the input feature is sequentially fed into three D2Conv layers with increased kernel sizes. Each convolution stage is equipped with residual connections to facilitate gradient flow and preserve original information. The multi-scale outputs are then aggregated via a lightweight Spatially-varying Aggregation (SA) module, inspired by spatial attention mechanisms (Li et al., 2025). The SA module uses a $1 \times 1$ convolution followed by a softmax function to generate spatial attention maps corresponding to each scale. Each attention map is multiplied element-wise with one of the three convolutional feature maps, dynamically recalibrating these features in a spatially adaptive manner. This simple design further enhances the dynamic capacity of ParaX with negligible parameter overhead, leading to more powerful feature representations. In summary, the key difference between our dynamic spatial mixing and Mona lies in the former being an input-dependent approach that, under PEFT conditions with limited learnable parameters, dynamically generates input-specific weights to better adapt to input variations and capture multi-scale contexts, whereas the latter employs input-independent weights, which are fixed and shared across all samples after training, resulting in limited representation capacity. Due to space constraints, additional analysis and discussions concerning the core idea of our method are provided in the Appendix.

## 4. Experiments

In this section, we present comprehensive experimental evaluations on various vision tasks, including semantic segmentation, object detection, instance segmentation, panop-

*Table 1.* Comparison of semantic segmentation performance on the ADE20K dataset. # P denotes the number of tunable parameters. Since all methods employ the same segmentation head, only the number of trainable parameters in the backbone network are reported.

| Method | Swin-B | | Swin-L | |
|---|---|---|---|---|
| | # P (M) | mIoU | # P (M) | mIoU |
| Full-tuning | 86.8 | 50.2 | 195.0 | 51.2 |
| VPT | 0.1 | 48.0 | 0.2 | 49.9 |
| LoRA | 5.4 | 49.4 | 8.1 | 51.1 |
| AdaptFormer | 5.4 | 50.0 | 8.1 | 51.3 |
| RepAdapter | 5.8 | 50.2 | 8.7 | 51.0 |
| LoRand | 5.9 | 49.9 | 8.2 | 51.4 |
| SNELL | 5.8 | 49.7 | 8.7 | 51.2 |
| Mona | 5.2 | 49.8 | 7.5 | 51.6 |
| **ParaX** | 5.2 | **50.3** | 7.3 | **52.0** |

| Method | ConvNeXt-B | | ConvNeXt-L | |
|---|---|---|---|---|
| | # P (M) | mIoU | # P (M) | mIoU |
| Full-tuning | 87.6 | 51.4 | 196.2 | 52.4 |
| LoRA | 5.9 | 50.5 | 8.8 | 50.8 |
| AdaptFormer | 6.4 | 50.3 | 9.6 | 50.9 |
| RepAdapter | 6.1 | 50.4 | 9.1 | 51.0 |
| LoRand | 6.7 | 49.6 | 9.3 | 51.2 |
| SNELL | 6.4 | 49.7 | 9.5 | 50.9 |
| CoLoRA | 6.4 | 50.8 | 9.4 | 51.2 |
| Mona | 6.5 | 50.7 | 9.1 | 51.5 |
| **ParaX** | 6.5 | **51.1** | 9.2 | **52.0** |

tic segmentation, and image classification. Most previous visual PEFT methods primarily focus on the image classification task (Mai et al., 2025). In contrast, we argue that parameter-efficient dense prediction is a more challenging and practically relevant problem, with numerous real-world applications. Hence, this work focuses primarily on dense prediction tasks. All experiments are conducted on 4 NVIDIA H800 GPUs.

**Pre-trained Models and Baselines**. We employ two representative hierarchical vision backbone networks for dense predictions, including Swin and ConvNeXt. In particular, we use the base and large versions pre-trained on ImageNet-21K with $224 \times 224$ inputs for both backbone architectures. Regarding image classification, we employ ViT-B/16 (Dosovitskiy et al., 2021) pre-trained with MAE (He et al., 2022), adhering to the setting in (Chen et al., 2022). This setup allows us to comprehensively validate the generalization ability and robustness of our method. Meanwhile, we also measure the performance of other representative baseline methods, including VPT (Jia et al., 2022), LoRA (Hu et al., 2022), AdaptFormer (Chen et al., 2022), RepAdapter (Luo et al., 2023), LoRand (Yin et al., 2023), SPT (He et al., 2023a), GPS (Zhang et al., 2024), SNELL (Shen et al., 2024), CoLoRA (Ran et al., 2025), and Mona (Yin et al., 2025), on the same tasks using the same pre-trained backbones. For a fair comparison, we adjust the latent dimension of adapter-based methods to ensure that they have a comparable number of trainable parameters. Note that we

*Table 2.* Comparison of object detection and instance segmentation performance on the COCO2017 dataset.

| Method | | Swin-B | | | | | | | Swin-L | | | | | |
|---|---|---|---|---|---|---|---|---|---|---|---|---|---|---|
| | # P (M) | $AP^b$ | $AP^b_{50}$ | $AP^b_{75}$ | $AP^m$ | $AP^m_{50}$ | $AP^m_{75}$ | # P (M) | $AP^b$ | $AP^b_{50}$ | $AP^b_{75}$ | $AP^m$ | $AP^m_{50}$ | $AP^m_{75}$ |
| Full-tuning | 86.8 | 47.5 | 69.8 | 52.3 | 42.8 | 66.6 | 46.0 | 195.0 | 48.6 | 70.9 | 53.7 | 43.8 | 68.1 | 47.3 |
| VPT | 0.1 | 40.6 | 65.7 | 44.0 | 38.8 | 62.7 | 41.3 | 0.2 | 42.6 | 67.8 | 46.1 | 40.5 | 64.9 | 43.1 |
| LoRA | 5.4 | 40.1 | 65.1 | 43.2 | 38.5 | 62.1 | 41.0 | 8.1 | 42.3 | 67.6 | 46.2 | 40.4 | 64.6 | 43.5 |
| AdaptFormer | 5.4 | 43.9 | 67.8 | 48.0 | 40.8 | 65.0 | 43.8 | 8.1 | 46.3 | 70.2 | 50.9 | 42.8 | 67.0 | 46.0 |
| RepAdapter | 5.8 | 44.7 | 68.3 | 48.9 | 41.4 | 65.1 | 44.6 | 8.7 | 46.9 | 70.5 | 51.6 | 43.1 | 67.3 | 46.7 |
| LoRand | 5.9 | 42.8 | 67.0 | 46.5 | 40.2 | 64.0 | 43.0 | 8.2 | 44.9 | 69.2 | 49.2 | 41.8 | 66.1 | 45.0 |
| SNELL | 5.8 | 41.3 | 66.3 | 44.6 | 39.3 | 63.1 | 42.0 | 8.7 | 43.8 | 68.4 | 47.8 | 41.2 | 65.2 | 44.3 |
| Mona | 5.2 | 46.6 | 69.4 | 50.9 | 42.4 | 66.2 | 45.6 | 7.5 | 48.1 | **71.3** | 52.8 | 43.9 | 68.2 | 47.6 |
| **ParaX** | 5.2 | **47.3** | **70.0** | **51.4** | **42.7** | **66.7** | **46.2** | 7.3 | **48.6** | 71.1 | **53.4** | **44.0** | **68.4** | **47.6** |

| Method | | ConvNeXt-B | | | | | | | ConvNeXt-L | | | | | |
|---|---|---|---|---|---|---|---|---|---|---|---|---|---|---|
| | # P (M) | $AP^b$ | $AP^b_{50}$ | $AP^b_{75}$ | $AP^m$ | $AP^m_{50}$ | $AP^m_{75}$ | # P (M) | $AP^b$ | $AP^b_{50}$ | $AP^b_{75}$ | $AP^m$ | $AP^m_{50}$ | $AP^m_{75}$ |
| Full-tuning | 87.6 | 47.8 | 69.7 | 52.4 | 43.0 | 66.9 | 46.3 | 196.2 | 48.1 | 69.7 | 53.1 | 43.2 | 66.8 | 46.7 |
| LoRA | 5.9 | 46.0 | 69.1 | 50.5 | 42.3 | 66.2 | 45.7 | 8.8 | 47.3 | 70.5 | 52.5 | 43.3 | 67.2 | 46.8 |
| AdaptFormer | 6.4 | 44.8 | 68.4 | 49.0 | 41.6 | 65.4 | 44.7 | 9.6 | 45.8 | 69.7 | 50.1 | 42.5 | 66.4 | 45.8 |
| RepAdapter | 6.1 | 46.1 | 68.9 | 50.9 | 42.3 | 65.9 | 45.8 | 9.1 | 46.8 | 69.6 | 51.6 | 42.3 | 66.8 | 46.3 |
| LoRand | 6.7 | 43.9 | 67.6 | 47.7 | 41.0 | 64.6 | 44.4 | 9.3 | 45.1 | 68.8 | 49.4 | 42.0 | 65.8 | 45.2 |
| SNELL | 6.4 | 43.5 | 67.3 | 47.0 | 40.7 | 64.1 | 43.7 | 9.5 | 44.6 | 68.3 | 48.7 | 41.1 | 65.1 | 44.3 |
| CoLoRA | 6.4 | 47.4 | 69.9 | 52.4 | 43.2 | 67.2 | 46.7 | 9.4 | 47.9 | 70.7 | 52.9 | 43.8 | 67.8 | 47.5 |
| Mona | 6.5 | 47.5 | 70.0 | 52.2 | 43.2 | 67.0 | 46.7 | 9.1 | 48.9 | 71.4 | 53.7 | 44.4 | 68.4 | 48.1 |
| **ParaX** | 6.5 | **48.0** | **70.2** | **52.7** | **43.5** | **67.5** | **46.9** | 9.2 | **49.5** | **71.9** | **54.4** | **44.8** | **69.2** | **48.4** |

retain the original configuration of VPT, as introducing additional prompt tokens would incur substantial computational overhead due to self-attention operations. Consequently, VPT has a lower parameter count compared to other methods. Meanwhile, since VPT is a specialized method for transformer adaptation, it is not applicable to ConvNeXt. Similarly, CoLoRA is not implemented for transformers, as it is tailored to ConvNets. Additionally, since SPT and GPS do not provide official implementations adapted to hierarchical architectures, we only report their classification performance in the main text, while the dense prediction performance comparisons based on ViT are provided in the Appendix due to page limits.

## 4.1. Semantic Segmentation

**Setup**. Semantic segmentation experiments are conducted on the ADE20K dataset (Zhou et al., 2017) using the UperNet framework (Xiao et al., 2018b). We adhere to the experimental setting in Swin (Liu et al., 2021), where all models are trained for 160K iterations using the AdamW optimizer (Loshchilov & Hutter, 2019) with a "poly" learning rate schedule (Chen et al., 2017) and a batch size of 16.

**Results**. Table 1 shows that our method achieves leading performance in semantic segmentation compared to the baselines. Specifically, using Swin-B as the backbone, ParaX achieves the highest mIoU of 50.3%, which is slightly better than the performance of full fine-tuning, while saving approximately 95% of the trainable parameters. Meanwhile, when using Swin-L as the backbone, ParaX achieves a notable performance improvement of 0.8% over full fine-

tuning while only using less than 4% of the parameters. Furthermore, compared to Mona, ParaX achieves performance increases of 0.5% and 0.4% using Swin-B and Swin-L, respectively. ParaX also significantly surpasses LoRand by 1.5% and 0.8% in mIoU when using ConvNeXt-B and ConvNeXt-L, respectively, and performs on par with full fine-tuning while using less than 8% of the parameters. On the other hand, we have also provided additional efficiency analysis in Appendix B.5, which demonstrates that our method achieves competitive computational efficiency in terms of throughput and GPU memory cost.

## 4.2. Object Detection and Instance Segmentation

**Setup**. Performance in object detection and instance segmentation is evaluated on the COCO2017 dataset (Lin et al., 2014) using the Mask R-CNN framework (He et al., 2017), following the same experimental setting in previous works (Liu et al., 2021; Lou & Yu, 2025). Specifically, all models are trained for 12 epochs using the AdamW optimizer with a batch size of 16.

**Results**. Unlike semantic segmentation, object detection and instance segmentation are more challenging, since models are required not only to predict bounding boxes but also to output instance-level segmentation masks, which test the spatial modeling ability of PEFT methods more directly. As shown in Table 2, our method delivers impressive performance compared to other PEFT methods. For instance, using Swin-B, ParaX improves over Mona by 0.7%/0.3% in $AP^b$/$AP^m$. When using Swin-L, ParaX achieves remarkable performance improvements of 3.7%/2.8% in $AP^b$/$AP^m$

*Table 3.* Comparison of panoptic segmentation performance on the COCO2017 dataset.

| Method | Swin-B | | | | Swin-L | | | |
|---|---|---|---|---|---|---|---|---|
| | # P (M) | PQ | SQ | RQ | # P (M) | PQ | SQ | RQ |
| Full-tuning | 86.8 | 50.3 | 81.3 | 60.6 | 195.0 | 51.4 | 81.5 | 61.9 |
| VPT | 0.1 | 45.1 | 78.6 | 55.4 | 0.2 | 46.6 | 79.1 | 57.0 |
| LoRA | 5.4 | 45.3 | 78.5 | 55.7 | 8.1 | 46.6 | 79.4 | 57.1 |
| AdaptFormer | 5.4 | 47.1 | 79.4 | 57.4 | 8.1 | 48.8 | 79.9 | 59.2 |
| RepAdapter | 5.8 | 47.9 | 79.9 | 58.1 | 8.7 | 49.2 | 80.7 | 59.6 |
| LoRand | 5.9 | 46.4 | 79.3 | 56.7 | 8.2 | 47.7 | 80.3 | 58.0 |
| SNELL | 5.8 | 46.0 | 79.2 | 56.4 | 8.7 | 47.2 | 79.4 | 57.6 |
| Mona | 5.2 | 48.1 | 79.9 | 58.3 | 7.5 | 49.7 | 80.7 | 60.2 |
| **ParaX** | 5.2 | **48.8** | **80.8** | **59.0** | 7.3 | **50.2** | **81.3** | **60.5** |

| Method | ConvNeXt-B | | | | ConvNeXt-L | | | |
|---|---|---|---|---|---|---|---|---|
| | # P (M) | PQ | SQ | RQ | # P (M) | PQ | SQ | RQ |
| Full-tuning | 87.6 | 50.2 | 81.2 | 60.5 | 196.2 | 51.0 | 80.9 | 61.4 |
| LoRA | 5.9 | 46.0 | 78.9 | 56.0 | 8.8 | 47.8 | 80.2 | 57.8 |
| AdaptFormer | 6.4 | 46.6 | 78.8 | 56.7 | 9.6 | 47.2 | 79.6 | 57.3 |
| RepAdapter | 6.1 | 47.1 | 79.7 | 57.2 | 9.1 | 47.8 | 79.9 | 57.9 |
| LoRand | 6.7 | 45.9 | 78.6 | 56.0 | 9.3 | 46.8 | 79.4 | 56.9 |
| SNELL | 6.4 | 45.5 | 78.9 | 55.5 | 9.5 | 46.9 | 79.2 | 57.1 |
| CoLoRA | 6.4 | 47.6 | 79.6 | 57.9 | 9.4 | 49.3 | 80.5 | 59.6 |
| Mona | 6.5 | 48.3 | 80.1 | 58.5 | 9.1 | 49.5 | 80.9 | 59.7 |
| **ParaX** | 6.5 | **48.9** | **81.0** | **59.1** | 9.2 | **50.4** | **81.2** | **60.7** |

over LoRand while performing on par with full fine-tuning. When using ConvNeXt, ParaX exceeds all other PEFT methods and full fine-tuning. Specifically, ParaX improves over full fine-tuning by 0.2%/0.5% in $AP^b/AP^m$ using ConvNeXt-B. When using ConvNeXt-L, ParaX significantly outperforms full fine-tuning by 1.4%/1.6% in $AP^b/AP^m$.

## 4.3. Panoptic Segmentation

**Setup**. We further perform evaluations on panoptic segmentation (Kirillov et al., 2019). This task unifies semantic segmentation, instance segmentation, and object detection, providing a more comprehensive evaluation of a model's dense prediction capability. Experiments are conducted on the COCO2017 dataset using the Panoptic FPN framework (Kirillov et al., 2019). All models are trained for 12 epochs using the AdamW optimizer with a batch size of 16. Performance metrics include panoptic quality (PQ), segmentation quality (SQ), and recognition quality (RQ).

**Results**. Table 3 provides the quantitative results of various models. Compared to other PEFT methods, our method delivers notable performance improvements. For instance, ParaX significantly surpasses AdaptFormer by 1.7% and 1.1% in PQ using Swin-B and Swin-L, respectively. Similarly, using ConvNeXt variants, ParaX clearly improves over CoLoRA by 1.3% and 1.1% in PQ. However, we notice that our method still has a moderate performance gap with full fine-tuning in this task. The reason might be that the extreme parameter efficiency of PEFT methods (less than 8% of the trainable parameters of full fine-tuning) imposes fundamental limits on their ability to capture the highly com-

plex and diverse features required for panoptic understanding. The concurrent demands of instance-level segmentation for things and semantic segmentation for stuff in panoptic segmentation necessitate a higher representation capability, which might not be fully attainable through PEFT alone. Nevertheless, our method still exceeds all baselines.

## 4.4. Image Classification

**Setup**. Following AdaptFormer (Chen et al., 2022), we conduct image classification experiments on three widely adopted datasets: CIFAR-100 (Krizhevsky et al., 2009), SVHN (Netzer et al., 2011), and Food-101 (Bossard et al., 2014). Additionally, we further conduct evaluations on the ImageNet-R dataset (Hendrycks et al., 2021), which exhibits a serious domain shift compared to ImageNet. In practice, for the first three datasets, we adopt the same data splits used in (Chen et al., 2022), while for ImageNet-R, we employ the data split as outlined in (Zhou et al., 2025), where 24,000 images are selected for training and the remaining 6,000 images for evaluations. The training setup follows the standard configuration as detailed in (Chen et al., 2022). Specifically, all models are trained for 100 epochs with an initial learning rate of 0.1, which is decayed using the cosine annealing schedule (Loshchilov & Hutter, 2017). The warm-up epochs and batch size are set to 20 and 1024, respectively.

**Results**. Table 4 reports the top-1 accuracy of different methods on each dataset and their average performance across the four datasets, revealing that our method delivers the best performance among all methods considered. In

*Table 4.* Comparison of image classification performance obtained using ViT-B/16.

| Method | # P (M) | CIFAR-100 | SVHN | Food-101 | ImageNet-R | Mean Acc. |
|---|---|---|---|---|---|---|
| Full-tuning | 86.0 | 85.9 | 97.7 | 90.1 | 64.0 | 84.4 |
| VPT | 0.1 | 82.4 | 94.0 | 83.0 | 65.6 | 81.3 |
| LoRA | 3.9 | 86.2 | 97.1 | 87.8 | 69.1 | 85.1 |
| AdaptFormer | 4.3 | 86.2 | 97.0 | 87.9 | 69.5 | 85.1 |
| LoRand | 3.9 | 86.1 | 96.9 | 87.9 | 69.7 | 85.1 |
| RepAdapter | 4.3 | 87.4 | 96.7 | 88.5 | 69.9 | 85.6 |
| SPT-LoRA | 3.7 | 86.5 | 97.2 | **89.8** | 72.6 | 86.5 |
| GPS | 3.5 | 86.8 | 97.4 | **89.8** | 72.4 | 86.6 |
| SNELL | 3.7 | 86.5 | 97.1 | 89.5 | 71.9 | 86.3 |
| Mona | 3.9 | 87.0 | 97.3 | 89.6 | 72.7 | 86.7 |
| **ParaX** | 3.9 | **87.7** | **97.7** | 89.7 | **74.3** | **87.4** |

*Table 5.* Impact of latent dimension and expert center capacity.

| Trade-off | # P (M) | $AP^b$ | $AP^m$ |
|---|---|---|---|
| $[M = 4L, \hat{C} = 40]$ | 5.5 | 46.1 | 42.0 |
| $[M = 2L, \hat{C} = 72]$ | 5.4 | 46.9 | 42.4 |
| $[M = L, \hat{C} = 128]$ | 5.2 | **47.3** | **42.7** |
| $[M = \frac{L}{2}, \hat{C} = 192]$ | 5.4 | 47.2 | 42.6 |

*Table 6.* Impact of expert center scope. Trainable parameter counts of all models are omitted due to negligible differences.

| # Layers/group | $AP^b$ | $AP^m$ |
|---|---|---|
| 18 | **47.3** | **42.7** |
| 9 | 47.1 | 42.5 |
| 6 | 46.5 | 42.0 |
| 3 | 46.3 | 41.9 |

comparison to the best-performing Mona, ParaX improves by 0.7%, 0.4%, and 0.1% in top-1 accuracy on the CIFAR-100, SVHN, and Food-101 datasets, respectively. This is likely because the decision boundaries in these datasets are relatively simple to learn. However, we observe that on the ImageNet-R dataset, our method exhibits a clear performance advantage over other competitors. For example, ParaX surpasses Mona by a notable 1.6% in top-1 accuracy. This indicates that even in the presence of a serious domain shift between the backbone's prior knowledge and the downstream data, ParaX can quickly learn to overcome such a shift using extremely few learnable parameters, owing to its powerful dynamic modeling capability. Since this paper primarily focuses on dense prediction tasks, we do not further explore classification performance due to resource limitations. However, we believe that the above results can demonstrate the superiority and robustness of our method in image classification.

## 4.5. Ablation Studies

**Setup**. We conduct comprehensive ablation studies on object detection and instance segmentation, utilizing Swin-B pre-trained on ImageNet-21K as the backbone network. The remaining experimental settings follow the configuration described in Section 4.2. Due to page limits, more experi-

mental evaluations are presented in the Appendix.

**Trade-off between Latent Dimension and Expert Center Capacity**. We investigate the effect of two key hyperparameters: the expert center capacity $M$ and the latent dimension $\hat{C}$. To enable easy integration of ParaX into various network architectures, we set $M$ according to the number of layers $L$ in each stage of the network. For example, in stage 3 of Swin-B, which has 18 layers, setting $M = 2L$ yields $M = 36$. Table 5 reveals that performance is governed by a balance between latent dimension and expert diversity. Optimal results occur at $[M = L, \hat{C} = 128]$, indicating that an overly small latent dimension or expert center capacity results in performance degradation.

**Effect of Shared Expert Center**. We evaluate the effectiveness of the shared expert center by varying its scope within the network. Specifically, in stage 3 of Swin-B (18 layers), we divide the expert center into smaller ones, each shared within a smaller group of consecutive layers. We experiment with groups of 3, 6, and 9 consecutive layers. Note that stages 1, 2, and 4 are not divided due to their small number of layers (2 layers only). As shown in Table 6, performance degrades as the groups become smaller. Notably, when expert centers are shared within small groups (e.g., 3 or 6 layers), the performance becomes comparable to Mona, indicating that the benefit of ParaX primarily comes from the proposed large shared expert center design.

## 5. Conclusion

In this paper, we propose a novel PEFT method for adapting vision models dubbed ParaX. Inspired by expert routing in MoE, our method constructs a large shared expert center where trainable parameter matrices serve as experts. Afterwards, a simple yet effective dynamic routing mechanism dynamically aggregates these experts to generate input-dependent projection weights for the network, thus improving cross-layer feature interaction and feature representation quality. Extensive evaluations on multiple challenging vision tasks demonstrate that our method achieves superior performance over existing visual PEFT methods.

## Impact Statement

This work proposes a generic transfer learning method for efficient visual recognition. All experiments are conducted on publicly available datasets and utilize open-source pre-trained models, thereby ensuring the absence of private or sensitive data. Furthermore, our research is designed to be domain-agnostic and poses no ethical concerns, as it is not specifically tailored towards potentially harmful application domains, such as surveillance or misinformation dissemination.

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

# Appendix

## A. More Experimental Comparisons

### A.1. Adapting ViT for Dense Prediction

**Setup**. Since some prior works primarily focus on adapting the ViT architecture (Jia et al., 2022; He et al., 2023a; Wang et al., 2024; Ren et al., 2025), we also adapt ViT using ParaX for a more comprehensive comparison with these methods. Specifically, following VPT (Jia et al., 2022), we conduct semantic segmentation experiments on ADE20K using the ViT-L model pre-trained on ImageNet-21K, with SETR-PUP (Zheng et al., 2021) chosen as the segmentation framework.

**Results**. Table 7 demonstrates that our method attains leading performance when adapting the ViT model, resulting in a significant improvement of 1.7% in mIoU compared with the best-performing baseline, demonstrating that ParaX can effectively adapt to the plain architecture for parameter-efficient dense predictions.

*Table 7.* Comparison of semantic segmentation performance using the ViT-L model.

| Method | # P (M) | mIoU |
|---|---|---|
| Full-tuning | 317.3 | 47.6 |
| VPT(Jia et al., 2022) | 13.6 | 44.1 |
| SPT-Adapter(He et al., 2023a) | 14.6 | 45.2 |
| SPT-LoRA(He et al., 2023a) | 14.6 | 45.4 |
| MLAE(Wang et al., 2024) | 14.6 | 45.0 |
| GPS(Zhang et al., 2024) | 14.9 | 45.8 |
| DA-VPT(Ren et al., 2025) | 13.6 | 45.1 |
| DA-VPT+(Ren et al., 2025) | 13.7 | 46.5 |
| **ParaX** | 14.7 | **48.2** |

*Table 8.* Comparison of semantic segmentation performance using the VMamba-B model.

| Method | # P (M) | mIoU |
|---|---|---|
| Full-tuning | 122.0 | 51.0 |
| AdaptFormer | 5.1 | 46.0 |
| RepAdapter | 5.5 | 46.4 |
| LoRand | 5.2 | 46.5 |
| SNELL | 5.0 | 46.1 |
| Mona | 4.6 | 47.5 |
| ParaX | 4.6 | **48.9** |

### A.2. Adapting Mamba-based Model

**Setup**. In addition to ConvNet- and Transformer-based models, Mamba (Gu & Dao, 2024) has recently demonstrated excellent performance in visual recognition (Liu et al., 2025a). Therefore, we also explore the adaptation of Mamba using our method. Specifically, we perform seman-

*Table 9.* Comparison of pose estimation performance using the Swin-B model.

| Method | P (M) | AP | AR |
|---|---|---|---|
| Full-tuning | 86.8 | 73.7 | 79.3 |
| VPT | 0.1 | 58.6 | 65.5 |
| LoRA | 5.4 | 50.2 | 61.5 |
| AdaptFormer | 5.4 | 57.8 | 64.6 |
| RepAdapter | 5.8 | 50.6 | 58.1 |
| LoRand | 5.9 | 58.3 | 65.0 |
| SNELL | 5.8 | 32.1 | 40.9 |
| Mona | 5.2 | 68.7 | 74.6 |
| **ParaX** | 5.2 | **70.5** | **76.3** |

*Table 10.* Comparison of remote sensing image segmentation performance.

| Method | # P (M) | Potsdam mIoU | LoveDA mIoU |
|---|---|---|---|
| Full-tuning | 86.8 | 79.3 | 54.8 |
| VPT | 0.1 | 77.2 | 52.4 |
| LoRA | 5.4 | 77.4 | 52.1 |
| AdaptFormer | 5.4 | 78.1 | 52.8 |
| RepAdapter | 5.8 | 78.2 | 52.6 |
| LoRand | 5.9 | 78.2 | 53.0 |
| SNELL | 5.8 | 77.9 | 52.9 |
| Mona | 5.2 | 78.6 | 53.8 |
| **ParaX** | 5.2 | **79.3** | **54.3** |

tic segmentation using the VMamba-B (Liu et al., 2024b) model pre-trained on ImageNet-1K, while using completely the same experimental setting as mentioned in Section 4.1.

**Results**. As shown in Table 8, when adapting VMamba, ParaX consistently demonstrates leading performance compared to other competitors, yielding a notable 1.4% improvement in mIoU over the second-best method. This validates the ability of our method to effectively adapt representative vision models.

### A.3. Human Pose Estimation

**Setup**. We investigate the efficacy of our method in the human pose estimation task using the COCO2017 dataset. During training, the Swin-B model pre-trained on ImageNet-21K is chosen as the backbone network, while the other experimental settings follow HRFormer (Yuan et al., 2021), including the use of a simple top-down head (Xiao et al., 2018a) and an input size of $256 \times 192$ pixels.

**Results**. In Table 9, it can be seen that there is a significant performance gap between previous PEFT works and full fine-tuning. Unlike segmentation and detection tasks, where the performance gap is relatively small (e.g., there is only a 3% difference in PQ between AdaptFormer and fine-tuning in panoptic segmentation), the gap in pose estimation is substantial, often exceeding 10% in AP. We attribute this

discrepancy to the fundamental difference in the granularity of task-specific knowledge acquired beyond the pre-training stage. Specifically, segmentation and detection tasks primarily involve region-level recognition, which is well aligned with the pre-trained weights obtained through image classification. Previous studies (Selvaraju et al., 2020; Chefer et al., 2021) have demonstrated that models can learn coarse region-level perception in image classification, where only image-level labels are available, making it easier for PEFT methods to adapt to segmentation and detection tasks. In contrast, pose estimation requires part-level geometric understanding (e.g., kinematic structure of the human body), which is largely absent in the objective of image classification. Therefore, the pose estimation task necessitates deeper feature rewiring for the backbone network, resulting in PEFT struggling to provide sufficient adaptation for such geometric perception. Nevertheless, our method significantly surpasses other PEFT methods, driven by input-dependent modeling and cross-layer feature communications.

### A.4. Remote Sensing Image Segmentation

**Setup**. Remote sensing image segmentation holds significant importance in real-world applications. To this end, we conducted experiments on two commonly used remote sensing segmentation datasets, including ISPRS Potsdam and LoveDA (Wang et al., 2021), using the Swin-B model pre-trained on ImageNet-21K, while the remaining experimental settings follow Section 4.1.

**Results**. As shown in Table 10, our method achieves excellent results in the remote sensing domain, demonstrating the best performance on both datasets. Specifically, ParaX improves over the best-performing baseline by 0.7% and 0.5% in mIoU on the two datasets, respectively. This notable improvement validates the generalizability of our method.

## B. More Ablation Studies

Building on the training settings outlined in Section 4.5, we present additional ablation experiments to systematically dissect the contribution of each component in our proposed method.

### B.1. Effect of Kernel Sizes in Multi-scale Spatial Mixing

We investigate the effect of kernel sizes of depthwise convolutions in Section 3.2. As shown in Table 11, using a single kernel size results in suboptimal performance, as it fails to capture object contexts at different scales. In contrast, using multiple kernel sizes yields better results. Specifically, the combination of kernel sizes $[3, 5, 7]$ achieves the best performance. However, further increasing the kernel sizes does not lead to better performance, which we attribute to the difficulty of learning with large kernel sizes under low-

*Table 11.* Effect of kernel sizes in multi-scale spatial mixing.

| Kernel Size | # P (M) | $AP^b$ | $AP^m$ |
|---|---|---|---|
| 3 | 5.1 | 46.3 | 42.2 |
| 5 | 5.2 | 46.4 | 42.2 |
| 7 | 5.4 | 46.6 | 42.5 |
| $[3, 5, 7]$ | 5.2 | 47.1 | 42.6 |
| $[5, 7, 9]$ | 5.5 | 46.9 | 42.6 |
| $[3, 5, 7]$ + SA | 5.2 | **47.3** | **42.7** |
| $[5, 7, 9]$ + SA | 5.5 | 47.2 | **42.7** |

rank conditions. Furthermore, incorporating the spatially varying aggregation (SA) module leads to modest performance improvements with a negligible increase in trainable parameters.

### B.2. Impact of Different Weight Initialization Strategies

By default, we adopt `trunc_normal` as the parameter initialization strategy for the expert centers. To further investigate the impact of different initialization methods, we also evaluate `kaiming_normal` and `kaiming_uniform`. As shown in Table 12, results demonstrate that our method is not sensitive to the choice of parameter initialization, that is, different strategies lead to only negligible differences in final performance, which confirms the training robustness of our approach.

*Table 12.* Impact of different parameter initialization methods.

| Init. Method | $AP^b$ | $AP^m$ |
|---|---|---|
| `trunc_normal` | **47.3** | **42.7** |
| `kaiming_normal` | 47.1 | 42.6 |
| `kaiming_uniform` | 47.2 | 42.7 |

### B.3. Sparse Expert Activations

Instead of incorporating all experts in the shared expert center at each layer, we sparsely activate the top-$K$ experts with the largest values from the gating vectors generated by the router network. Specifically, in stage 3 of Swin-B (18 layers), we set $K = \{3, 6, 12\}$. As shown in Table 13, the sparse activation strategy leads to performance degradation, while activating all experts simultaneously achieves the best results. We suggest two possible reasons for this phenomenon: (1) Dense prediction tasks are inherently complex, and each expert has relatively few parameters, thus requiring the integration of knowledge from multiple experts to enhance representations; (2) To ensure computational efficiency, our router network is relatively simple and does not incorporate sophisticated gating mechanisms. Although sparse activation is commonly used in MoE-based Large Language Models (LLMs) to reduce computational costs (Rajbhandari et al., 2022; Dai et al., 2024; Cai et al., 2025), the main contribution of this work lies in the shared expert

center, rather than designing complex MoE architectures. Nevertheless, exploring how to effectively integrate sparse activation strategies into our method is a potentially promising direction for future work.

Table 13. Effect of sparse expert activations.

| Top-$K$ | AP$^b$ | AP$^m$ |
|---|---|---|
| $K$=3 | 46.3 | 41.8 |
| $K$=6 | 46.6 | 42.6 |
| $K$=12 | 47.0 | 42.6 |
| $K$=18 (All) | **47.4** | **42.8** |

### B.4. More Design Choices

We further examine several design choices in ParaX. First, we evaluate the layout of multi-scale dynamic convolutions introduced in Section 3.2. As shown in Table 14 (a), our sequential layout of multi-kernel convolutions with residual connections yields better performance compared to both a residual-free sequential layout and a parallel layout without introducing extra trainable parameters.

Then, we test the activation functions used in the router. Table 14 (b) indicates that employing both ReLU and Sigmoid activations results in performance degradation, as it does not adequately model the competition among experts in the shared center. In contrast, softmax activation provides a better probability distribution over experts.

Finally, Table 14 (c) demonstrates the impact of the hidden channel size in the router network of ParaX on performance. It can be observed that excessively small hidden channel dimensions lead to performance degradation, whereas overly large dimensions increase the number of parameters with negligible performance gains. Therefore, we select an appropriate intermediate value.

### B.5. Computational Efficiency Analysis

To evaluate computational efficiency, we measure training throughput (Thr.) and GPU memory usage (Mem.) using Swin-L as the backbone. We focus solely on the efficiency of the backbone network since PEFT modules are only integrated into the backbone network. Specifically, the input size is set to 1280×800 with a batch size of 4, and measurements are averaged over more than 100 iterations using a single NVIDIA L40S GPU based on the COCO2017 dataset.

As shown in Table 15, our method achieves a favorable trade-off between computational efficiency and accuracy. In particular, it matches the speed and memory footprint of the best-performing baseline while delivering clearly better performance. Although our approach is approximately 15% slower than full fine-tuning during training, this overhead mainly stems from the use of multi-kernel convolutions

Table 14. Impact of other design choices.

*(a)* Effect of the layout of multi-kernel dynamic convolutions.

| Method | AP$^b$ | AP$^m$ |
|---|---|---|
| Parallel | 47.1 | 42.6 |
| Sequential (w/o Res.) | 46.6 | 42.3 |
| Sequential (w Res.) | **47.3** | **42.7** |

*(b)* Effect of activation function in dynamic routing.

| Activation | AP$^b$ | AP$^m$ |
|---|---|---|
| ReLU | 45.8 | 41.5 |
| Sigmoid | 46.2 | 41.9 |
| Softmax | **47.3** | **42.7** |

*(c)* Effect of different hidden channel sizes in router network.

| Hidden Size | # P (M) | AP$^b$ | AP$^m$ |
|---|---|---|---|
| 8 | 5.0 | 47.0 | 42.5 |
| 16 | 5.2 | 47.3 | 42.7 |
| 40 | 5.9 | **47.4** | **42.8** |

Table 15. Comparison of efficiency at the input resolution of 1280×800 with a batch size of 4.

| Method | # P (M) | Thr. (imgs/s) | Mem. (GB) | AP$^b$ | AP$^m$ |
|---|---|---|---|---|---|
| Full-tuning | 195.0 | 4.6 | 3.2 | 48.6 | 43.8 |
| LoRA | 8.1 | 4.1 | 2.4 | 42.3 | 40.4 |
| AdaptFormer | 8.1 | 5.2 | 2.1 | 46.3 | 42.8 |
| RepAdapter | 8.7 | 4.8 | 2.5 | 46.9 | 43.1 |
| LoRand | 8.2 | 5.1 | 2.5 | 44.9 | 41.8 |
| SNELL | 7.5 | 5.0 | 2.3 | 41.3 | 39.3 |
| Mona | 7.5 | 3.9 | 3.3 | 48.1 | 43.9 |
| **ParaX$^‡$** | 7.0 | 5.0 | 2.7 | 47.7 | 43.5 |
| **ParaX** | 7.3 | 4.0 | 3.2 | 48.6 | 44.0 |

in the adapter design, following Mona. To analyze this overhead, we construct a simplified variant, termed ParaX$^‡$, which removes all multi-kernel convolutions. This variant achieves throughput and memory usage comparable to other PEFT methods. For example, compared to AdaptFormer, ParaX$^‡$ yields a significant improvement of 1.4%/0.7% in AP$^b$/AP$^m$ with comparable throughput. Meanwhile, our method surpasses LoRand by a large margin of 2.8%/1.7% in AP$^b$/AP$^m$ with similar GPU memory costs. These results further indicate that, although the shared expert center enables reusing a small set of learnable parameters across different network layers, it introduces only negligible additional computations, demonstrating competitive computational efficiency.

On the other hand, full fine-tuning requires storing a separate version of the network parameters for each downstream task, while each version is nearly as large as the original pre-trained model. This leads to significant storage overhead

when multiple tasks are carried out. In contrast, ParaX requires less than 4% of the trainable parameters to achieve comparable performance, reducing the storage cost by more than 95%.

# C. More Analysis and Discussions

## C.1. Expert Allocation Pattern Analysis

To understand the expert-allocation behavior of our method, we conduct a visualization study using a Swin-B model trained on COCO2017 (Section 4.2), focusing on Stage 3, which exhibits the optimal layer-wise diversity. We visualize the router softmax scores in ParaX (inserted after the FFN), which are used to generate the dynamic matrices $\mathbf{W}_1$ and $\mathbf{W}_2$ (Section 3.2). Specifically, the activation maps are obtained by averaging the scores over various numbers of input images randomly sampled from the COCO2017 validation set. As shown in Figure 5, the expert allocation behaviors differ significantly between $\mathbf{W}_1$ and $\mathbf{W}_2$. For $\mathbf{W}_1$, the router produces a highly peaked distribution, strongly favoring a single expert per layer. In contrast, the activation for $\mathbf{W}_2$ is more distributed, with scores spread across multiple experts, and this pattern varies across layers. We attribute this disparity to the fundamental nature of their respective learning objectives. The $\mathbf{W}_1$ matrix, responsible for compressing information by significantly reducing channels, necessitates preservation of feature compression, prompting the router to identify and leverage a minimal set of the most relevant experts for maximal information preservation. Conversely, the $\mathbf{W}_2$ matrix, which entails reconstructing information by expanding smaller channels to larger ones, benefits from a consensus of multiple related experts, thereby facilitating the effective interpretation and inversion of the compressed representations.

On the other hand, the expert activation maps also exhibit differences when using different numbers of input images, which arises from the dynamic parameter routing mechanism, i.e., different images result in different router softmax scores, thereby generating different $\mathbf{W}_1$ and $\mathbf{W}_2$ matrices for input-dependent feature modeling. This is one of the key factors that enable ParaX to produce more expressive feature representations.

## C.2. Implicit Cross-layer Interaction

We provide a theoretical explanation for why ParaX enables stronger cross-layer interaction than previous PEFT methods. For simplicity, suppose a model with $L$ layers in total, let $\mathcal{L}$ be the loss function, and let $h_l$ denote the output feature at layer $l$. In conventional PEFT methods (e.g., AdaptFormer), the gradient with respect to the trainable weight matrix $W_l$ is isolated within layer $l$:

$$\frac{\partial \mathcal{L}}{\partial W_l} = \frac{\partial \mathcal{L}}{\partial h_l} \cdot \frac{\partial h_l}{\partial W_l}, \tag{1}$$

where $\frac{\partial \mathcal{L}}{\partial h_l}$ represents the back-propagated gradients arriving at layer $l$, and $\frac{\partial h_l}{\partial W_l}$ denotes the local gradient at layer $l$. This indicates that the gradient $\frac{\partial \mathcal{L}}{\partial W_l}$ does not explicitly account for local gradients from other layers. In this case, while a gradient update at an earlier layer (e.g., $l = 1$) can directly alter its own feature representation (output feature of the current layer), its effect on deeper layers (e.g., $l = 10$) must propagate sequentially through the network's forward pass.

In contrast, each effective weight matrix $W_l$ in ParaX is constructed via a shared expert center (assuming there are $M$ learnable experts) termed as $E = \{\mathcal{E}_1, \mathcal{E}_2, \ldots, \mathcal{E}_M\}$:

$$W_l = \sum_{m=1}^{M} g_l^m \mathcal{E}_m, \tag{2}$$

where $g_l^m$ is the routing coefficient for expert $m$ at layer $l$. Theoretically, a given trainable expert $\mathcal{E}_m$ can be utilized by every network layer, and thus the gradient with respect to $\mathcal{E}_m$ is calculated as:

$$\frac{\partial \mathcal{L}}{\partial \mathcal{E}_m} = \sum_{l=1}^{L} \frac{\partial \mathcal{L}}{\partial h_l} \cdot \frac{\partial h_l}{\partial W_l} \cdot \frac{\partial W_l}{\partial \mathcal{E}_m} \tag{3}$$

This provides a key distinction from other PEFT methods: the gradient for updating $\mathcal{E}_m$ is obtained by explicitly aggregating layer-wise contributions across all $L$ layers. Specifically, for each layer $l$, the contribution comprises: 1) The loss gradient back-propagated to the layer's output feature ($\partial \mathcal{L}/\partial h_l$); 2) The layer's local gradient of output to its weights ($\partial h_l/\partial W_l$). By applying the chain rule with $\partial W_l/\partial \mathcal{E}_m$ and summing these terms over $l = 1$ to $L$, the update of $\mathcal{E}_m$ holistically integrates gradient information flowing through each layer of the entire network. In this way, once $\mathcal{E}_m$ is updated, the weight matrices in all other layers that share $\mathcal{E}_m$ are immediately affected, thereby enabling more direct modulation of feature representations in each layer. Consequently, without relying on memory-costly explicit feature connections across different layers, we still establish an implicit cross-layer interaction by efficiently modeling complex information flows across different layers during training, thereby enhancing feature diversity compared to other PEFT methods (Figure 1), validating our insights in Section 1. It is noteworthy that this mechanism also constitutes a key difference between ParaX and other MoE-based PEFT methods.

In practice, although individual experts may not be activated by every layer (Figure 5), they are typically shared among

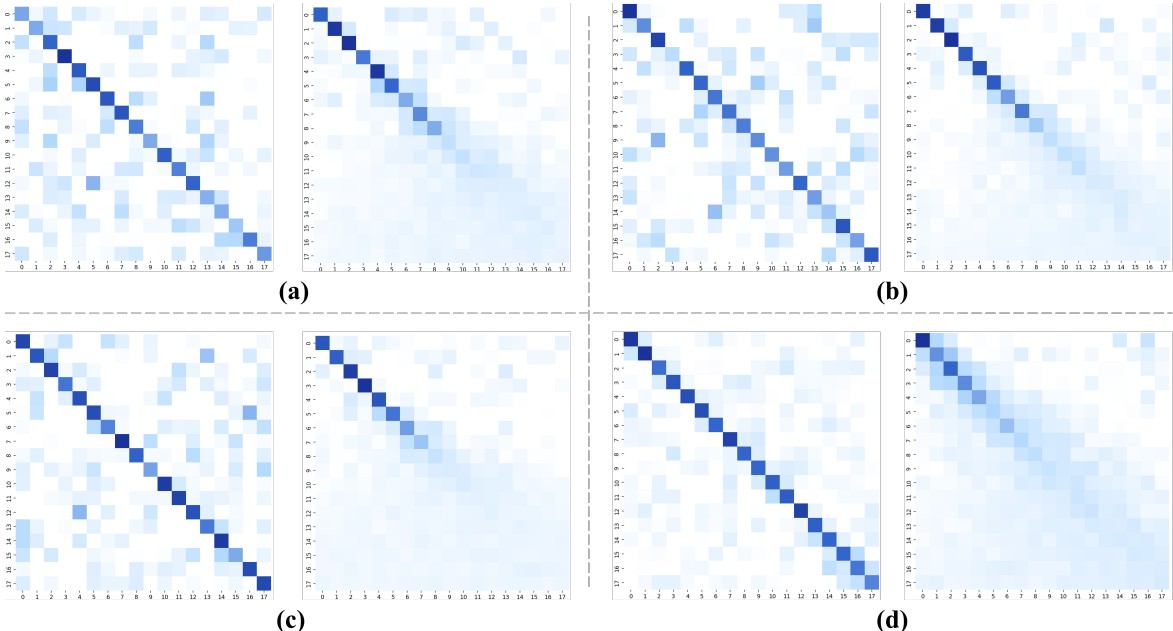

*Figure 5.* Expert activation maps in ParaX. Panels **(a)**-**(d)** are generated using 20, 40, 80, and 160 randomly selected images from the COCO2017 validation set, respectively. In each subfigure, the left and right heatmaps denote the expert activations for generating the channel-reducing matrix ($\mathbf{W}_1$) and the channel-expanding matrix ($\mathbf{W}_2$), respectively. The horizontal and vertical axes indicate the expert and the layer indices, respectively.

several layers, thereby inducing implicit cross-layer inter-action among those layers. Notably, the above theoretical analysis is consistent with our previous empirical results. In Table 6, enlarging the scope of the expert center consistently improves performance over smaller scopes, since a larger scope can result in stronger cross-layer interactions. Meanwhile, Table 13 shows that sparse expert activations can degrade performance, as it weakens these implicit cross-layer interactions.

### C.3. More Discussions with Related Methods

To provide a clearer understanding of the distinctions between our method and existing MoE-based PEFT approaches, we conduct a detailed technical comparison with two representative works: LoRand (Yin et al., 2023) and MLAE (Wang et al., 2024). Both methods explore expert allocation mechanisms for visual PEFT, but exhibit fundamental limitations in terms of input dependency and feature diversity.

Specifically, while LoRand constructs adapter parameters through a simple summation of multiple fixed parameter groups, and MLAE employs random expert masking strategies, both methods suffer from critical limitations in achieving input-dependent adaptation. Notably, the weight fusion mechanism in LoRand is not a weighted aggregation scheme, and cannot dynamically adjust the fusion weights of different parameter groups according to input features,

resulting in input-agnostic adaptation. Similarly, the stochastic masking strategy of MLAE operates independently of the input features, not able to customize expert activation patterns according to specific visual cues or task requirements. Furthermore, neither method addresses the issue of cross-layer feature redundancy. Without mechanisms to promote interaction between layers, PEFT methods tend to learn repetitive feature representations across network layers as aforementioned, limiting their representation capacity.

In contrast, ParaX makes two key contributions to address these limitations: shared expert centers across layers and a lightweight dynamic router that generates dynamic gating vectors. This allows our method to achieve not only input-dependent adaptation but also diversified features across layers, achieving strong visual adaptation in a wide range of challenging tasks. On the other hand, although there exist several MoE-based PEFT methods in the NLP domain (Luo et al., 2024; Tian et al., 2024; Zadouri et al., 2024; Dou et al., 2024; Gao et al., 2025), our approach introduces a fundamental distinction through the concept of shared expert centers

## D. Limitations and Future Works

Although ParaX achieves better performance than other PEFT methods in diverse vision tasks, and sometimes even surpasses full fine-tuning using an extremely reduced number of trainable parameters, it still has limitations. Like

representative vision adapters such as AdaptFormer and Mona, ParaX cannot be integrated into the original network during inference. In addition, the introduction of multi-scale depthwise convolutions results in a minor increase in training latency. Furthermore, due to resource constraints, we do not conduct evaluations on larger-scale models, such as DINOv3-ViT-7B (Siméoni et al., 2025). In our future work, we aim to further improve efficiency, reduce the number of trainable parameters, and achieve even stronger performance, while also expanding the evaluations to larger models. Moreover, it is also interesting to explore the extension of our methods to LLMs and Multimodal LLMs.

