# OpenReview forum: "Parameters as Experts: Adapting Vision Models with Dynamic Parameter Routing"
_ICML.cc/2026/Conference — ICML 2026 regular_

### Official Review · Reviewer_7skm · 2026-02-25

**Soundness:** 3
**Presentation:** 4
**Significance:** 3
**Originality:** 3
**Overall Recommendation:** 5
**Confidence:** 3

**Summary:**

The paper proposes AdaRoute, a new parameter-efficient fine-tuning (PEFT) method for adapting vision models. It introduces shared expert centers containing trainable parameter matrices. A dynamic parameter routing mechanism generates input-dependent weight matrices for each network layer by selectively combining these experts. The authors evaluate AdaRoute on semantic segmentation, object detection, instance segmentation, panoptic segmentation, and image classification.

**Compliance With Llm Reviewing Policy:**

Affirmed.

**Final Justification:**

My concerns are solved, so I raised my score to 5.

**Key Questions For Authors:**

The title explicitly mentions "dense predictions", but the proposed method and the experiments are not strictly limited to dense prediction tasks.  Would you consider revising the title to reflect this broader applicability?

Could you provide a simple ablation study on the input-dependent design? I am curious to see the performance if the dynamic coefficients are standard trainable parameters rather than outputs generated by the router network.

What is the diversity among the parameter experts after training is complete? It would be helpful to visualize the similarity matrix between the learned experts.

The AdaRoute method itself does not appear to be specific to vision tasks. How would this approach perform if applied to NLP models and tasks?

**Limitations:**

yes

**Strengths And Weaknesses:**

## Strengths:

- The motivation is clear and addresses the limitations of existing methods well. Figure 1 provides a very intuitive visualization that shows how the proposed method effectively solves these limitations.
- The method is sensible and logical. While concepts like input-dependent adaptation or MoE-based LoRA are not entirely new, the specific design of AdaRoute is well thought out and effective.
- The experiments are extremely thorough and convincing. The authors tested the method on various tasks and backbone models with good results. The paper also provides comprehensive metrics, including trainable parameters, GPU memory, and training throughput, rather than just reporting accuracy.
- The writing is clear and easy to follow.

## Weaknesses:
I do not have major criticisms of the technical work, but I have a few suggestions and questions regarding the scope and ablation studies, which are listed in the questions section below.

---

> ### Author Rebuttal · Authors · 2026-03-27
>
> >Q1: Title Improvement
>
> A1: We appreciate this thoughtful suggestion. Indeed, our method is not limited to dense prediction tasks, as demonstrated by various experimental evaluations. Removing "for Dense Predictions" from the title can better reflect the versatility, and we would be more than happy to further discuss this with the reviewer.
>
>
>
> >Q2: Input-independent Variant
>
> A2: Thank you for this insightful comment. We remove the router network from each layer and replace it with learnable parameters, followed by a softmax to produce coefficients. Once training is completed, the coefficients remain fixed, eliminating input-dependent modeling. For fair comparison, we increase the expert hidden rank to match the original parameter count. Experiments are conducted on COCO detection (Swin-B+Mask R-CNN) and ImageNet-R classification (ViT-B/16):
>
> | COCO| #P (M) | AP^b | AP^m |
> |---|---|---|---|
> | Dynamic | 5.2 | 47.3 | 42.7 |
> | w/o Dynamic | 5.2 | 46.7 | 42.4 |
>
> | IN-R | #P (M) | Top-1 |
> |---|---|---|
> | Dynamic | 5.2 | 74.3 |
> | w/o Dynamic | 5.2 | 73.1 |
>
> Removing dynamic routing causes a notable performance drop on both tasks. The model can no longer tailor expert aggregation to each input, limiting the expressiveness of the projection matrices. The drop is more notable on ImageNet-R (1.2 top-1): under a larger domain shift from pre-trained models, the router adaptively adjusts expert aggregation for unseen patterns, whereas fixed coefficients lack this flexibility. This ablation directly confirms the contribution of input-dependent routing (Sec.3.2). We will add related results and more discussions in the revision.
>
>
> >Q3: Expert Diversity Visualization
>
> A3: Thank you for this thoughtful comment. Based on the settings in Sec.C.1, we compute pairwise cosine similarity among expert matrices from $\mathcal{E}_A$ and show the results for a subset of expert matrices with indices (3, 6, 9, 18) below:
>
> | | $\mathcal{E}_3$ | $\mathcal{E}_6$ | $\mathcal{E}_9$ | $\mathcal{E}_{18}$ |
> |---|---|---|---|---|
> | $\mathcal{E}_3$ | 1.000 | 0.003 | 0.006 | 0.006 |
> | $\mathcal{E}_6$ | 0.003 | 1.000 | 0.010 | 0.013 |
> | $\mathcal{E}_9$ | 0.006 | 0.010 | 1.000 | 0.014 |
> | $\mathcal{E}_{18}$ | 0.006 | 0.013 | 0.014 | 1.000 |
>
> The mean pairwise cosine similarity between different experts is only 0.009 for the setting in Sec.C.1, with a maximum similarity of 0.021 (a complete version will be provided in the revised version). This confirms that experts capture diverse representations. Combined with Fig.5, where different layers exhibit distinct routing distributions, these findings demonstrate that AdaRoute can learn to promote expert specialization. The low CKA similarity in Fig.1(c) further corroborates this quantitative finding from the feature diversity perspective.
>
>
> >Q4: Adapt to NLP Tasks
>
> A4: Thank you for this thoughtful comment. To adapt AdaRoute to NLP, we remove multi-scale 2D convs (vision-specific) and retain the router network and shared expert center. Following HiRA (ICLR'25), we apply AdaRoute to all attention and FFN layers (q_proj, k_proj, v_proj, up_proj, down_proj) in each transformer block with 32 shared experts, rank 24, and a 2-layer MLP router. The total parameter count is adjusted to be comparable to baselines. We evaluate on 8 commonsense reasoning datasets using Llama3-8B:
>
> | Llama3-8B| P (%) | BoolQ | PIQA | SIQA | ARC-c | ARC-e | OBQA | HellaS | WinoG | Average |
> | :------------ | :--------- | :---- | :--- | :--- | :---- | :---- | :--- | :----- | :---- | :------ |
> | Prompt Tuning | 0.0010     | 56.9  | 45.1 | 36.1 | 31.6  | 32.7  | 29.2 | 14.0   | 50.1  | 37.0    |
> | P-Tuning      | 0.6240     | 60.0  | 11.6 | 8.2  | 7.4   | 8.6   | 9.6  | 1.8    | 37.7  | 18.1    |
> | LoRA          | 0.7002     | 70.8  | 85.2 | 79.9 | 71.2  | 84.2  | 79.0 | 91.7   | 84.3  | 80.8    |
> | DoRA          | 0.7002     | 74.6  | 89.3 | 79.9 | 80.4  | 90.5  | 85.8 | 95.5   | 85.6  | 85.2    |
> | MoRA          | 0.6997     | 74.3  | 87.4 | 80.7 | 79.6  | 91.2  | 85.6 | 43.5   | 86.7  | 78.6    |
> | HiRA          | 0.7002     | 75.4  | 89.7 | 81.2 | 82.9  | 93.3  | 88.3 | 95.4   | 87.7  | 86.7    |
> | Ours          | 0.7011     | 72.9  | 88.4 | 80.0 | 82.0  | 93.0  | 88.0 | 95.4   | 87.9  | 86.0    |
>
> AdaRoute achieves 86.0% average accuracy, substantially outperforming LoRA and MoRA while remaining competitive with HiRA, a method specifically designed and tuned for NLP adaptation. Notably, these results are achieved without any NLP-specific hyperparameter tuning, underscoring the generality of our core design. Due to limited computational resources, we have not yet tuned adaptation strategies specifically for NLP and will investigate this further in future work.
>
> We sincerely hope our responses have thoroughly addressed your concerns and would deeply appreciate it if you could increase your score in light of our revisions.

---

> > ### Author Rebuttal · Reviewer_7skm · 2026-03-31
> >
> > Thanks for the rebuttal. All my concerns are solved and really appreciate your efforts in doing those NLP related experiments. I will raise my score. Good luck.

---

> > > ### Author Response · Authors · 2026-04-05
> > >
> > > We sincerely appreciate the reviewer's decision to raise the score (``4-->5``). We will carefully incorporate the additional results and discussions into the revised manuscript to further strengthen the paper. Once again, many thanks for your meticulous review and insightful comments.

---

### Official Review · Reviewer_MaEV · 2026-02-26

**Soundness:** 3
**Presentation:** 3
**Significance:** 2
**Originality:** 2
**Overall Recommendation:** 5
**Confidence:** 3

**Summary:**

This work introduces AdaRoute, a method that integrates mixture-of-experts (MoE) in each stage. In each stage, a shared expert is used to dynamically generate low-rank matrixes for a downstream task. Extensive four visual tasks (semantic segmentation, object detection and instance segmentation, and panoptic segmentation) are used to demonstrate AdaRoute’s improved performance over baselines. Further ablation studies yield deeper insights.

**Compliance With Llm Reviewing Policy:**

Affirmed.

**Final Justification:**

This work shows improved performance over baselines with extensive experiments and medium novelty. The presentation is decent. The authors have addressed most of my clarifications during the rebuttal. I have raised my rating to __5: Accept__.

**Key Questions For Authors:**

__Q1__ (__OR_W1__): Since the expert centers are defined stage-wise (shared across layers within a stage), it would be helpful to clarify whether using a single large global expert center across all stages could provide additional benefits or trade-offs. Such design can extend layer interactions to various stages. Such experiments can help justify the stage-wise expert center design choice.

__Q2__ (__P_W7__): (Clarity) When computing # P (M) of AdaRoute, does it include all experts in all expert centers across stages, or just the final one (in each stage) that has already been multiplied by the dynamic coefficients? It would be great to provide such clarity in terms of M1, M2, C, $\hat{C}$ etc.

**Limitations:**

Yes but mainly limited to technical perspective. Some potential negative societal impact could be security or adversarial attack, such as attackers can manipulate the routing mechanism to influence the model's behavior.

**Strengths And Weaknesses:**

__Soundness (SN)__

__Strengths__:

__SN_S1__: Overall, several key vision tasks, including semantic segmentation, object detection and instance segmentation, panoptic segmentation and image classification, have been used to evaluate the effectiveness of the method, demonstrating the improved performance over recent methods with close # trainable parameters From Table 1 to 4.

__SN_S2__: It is good to use effective receptive field (ERF) to measure representation deficiency compared to existing works, thereby motivating the need for a larger receptive field that is closer to that of a fully fine-tuned model, especially in comparison to prior PEFT approaches that rely on fixed adapters.

__SN_S3__: It is great that this work employs Centered Kernel Alignment (CKA) analysis to systematically assess feature redundancy in comparison with existing approaches, thereby providing empirical evidence for the need to enhance representational richness beyond prior PEFT methods that rely on fixed adapters.

__Weaknesses__:

__SN_W1__: Most of the performance gains are not quite significant (e.g. less than 1% from Table 1-4).

__SN_W2__: The claim of cross-layer interaction is limited in one stage as Expert Centers are defined per stage.

__SN_W3__: __Unclear evidence of better spatial dependency capturing ability can improve visual dense task performace.__ This work claimed that a model’s ability to capture complex spatial dependencies (SD) is essential for dense prediction tasks. Although the ERF visualization in Figure 1(c) shows that the proposed method AdaRoute has a larger receptive field than the baselines, it remains unclear whether the performance gains in several dense vision tasks (Tables 1–3) can be directly attributed to improved spatial dependency (SD) modeling. By integrating mixture-of-experts (MoE) in this work, it also introduces dynamic parameter routing mechanism (RT) that is different from prior work (e.g. Mona). In addition, the CKA map in Figure 1. (c) and the main text argue that AdaRouter captures more diverse representations (DR) than prior work. For dense prediction tasks, spatial precision (e.g. accurate object boundary) modeling can be as important as capturing global context through larger spatial dependency (SD). All of these can contribute to the performance gain but the current form lacks such clarity. Therefore, it is encouraged to design cleaner experiments to support the claim of “better spatial dependency capturing ability” -> “better visual dense task performance.”

___

__Presentation (P)__

__Strengths__:

__P_S1__: Overall, this paper is well presented with details and clear logic flows. Most of the tables are figures are presented in proper size.

__Weaknesses__:

__P_W1__: (Clarity) In Fig. 1 (c) what does the input look like? Is it obtained by feeding just one sample from COCO2017 or the whole COCO dataset? Providing such clarity would help.

Similarly, CKA requires input data, as it measures the similarity between layer-wise representations (i.e., activations) produced in response to those inputs. It would be great to briefly provide the clarity of inputs.

__P_W3__: (Clarity) Are Fig. 1 (c) ERF and CKA in Figure 1 obtained on (1) solely adapters or (2) in the W_pre-trained + BA (Adapters) fashion?

__P_W4__: The overview is illustrated on Line 182 on Page 4 using Figure 2, but Figure 2 is on top of Page 3. It is generally advised to put them closer for better readability.

__P_W5__: (Minor) The digits in Fig. 5 are very small. They might not be legible when the paper is printed.

__P_W6__: The # P (M) of VPT is 13.6 in Table 7, which is significantly larger than VPT of 0.1 in any other tables. It would be great to briefly provide clarity on this large discrepancy and how they are reported.

__P_W7__: (Clarity) When computing # P (M) of AdaRoute, does it include all experts in all expert centers across stages, or just the final one (in each stage) that has already been multiplied by the dynamic coefficients? It would be great to provide such clarity in terms of M1, M2, C, $\hat{C}$ etc.

__P_W8__: It is not quite easy to connect Figure 1-4 to have a hierarchical understanding of the method. It might be beneficial to leave some common components between figures for better connectivity.

___

__Significance (SF)__

__Strengths__:

__SF_S1__: The simplicity of integrating MoE might encourage further usage.

__Weaknesses__:

__SF_W1__: __More experiments focused on domain shift might help distinguish AdaRoute.__ It is encouraging that AdaRoute surpasses Mona by a larger margin (1.6%) on ImageNet-R, whereas the gains on other tasks are typically below 1%. This suggests that AdaRoute may be particularly effective under domain shift. It would be valuable to further investigate this aspect and conduct additional experiments explicitly targeting domain shift, to better demonstrate AdaRoute’s advantages over the baselines and to evaluate whether it more effectively mitigates the gap between the pre-trained backbone and downstream tasks under distribution changes.

___
__Originality (OR)__

__Strengths__:

__OR_S1__: Although __OR_W1__, this paper integrates MoE into the low-rank adapters in its own fashion, and clearly demonstrating how it differs from prior work when introducing each component.

__Weaknesses__:

__OR_W1__: Since the expert centers are defined stage-wise (shared across layers within a stage), it would be helpful to clarify whether using a single large global expert center across all stages could provide additional benefits or trade-offs. Such design can extend layer interactions to various stages. Such experiments can help justify the stage-wise expert center design choice.

---

> ### Author Rebuttal · Authors · 2026-03-27
>
> >SNW1&SFW1: Significance of Performance Gains
>
> A1: Since all PEFT methods use <8% of full fine-tuning parameters, the performance ceiling is naturally compressed. Therefore, gains over Mona (~0.7%) in Tab.1-Tab.4 are meaningful, and many results exceed 1%:
>
> - Tab.9 Pose: +1.8 AP, +1.7 AR over Mona
> - Tab.3 Panoptic: +1.7 PQ over AdaptFormer (Swin-B)
> - Tab.7 ViT-L Seg: +1.7 mIoU over the best baseline
> - Tab.4 ImageNet-R: +1.6 top-1 over Mona
> - Tab.2 ConvNeXt-L Det: +1.4/1.6 AP^b/AP^m over full-tuning
> - Tab.8 VMamba Seg: +1.4 mIoU over Mona
>
> As you suggested, we further conduct domain adaptation (DA) on ``Cityscapes->Foggy Cityscapes`` segmentation using classical AdaptSegNet+Swin-B:
>
> | Method | #P (M) | mIoU |
> |---|---|---|
> | Full-tuning | 86.8 | 64.1 |
> | Mona | 5.2 | 61.9 |
> | AdaRoute | 5.2 | 63.6  |
>
> AdaRoute narrows the gap with full-tuning to 0.5 mIoU, while Mona trails by 2.2. Input-dependent routing dynamically adjusts expert aggregation for unseen distributions, while regular adapters lack this flexibility, aligning with the +1.6 ImageNet-R gain (Tab.4). These results effectively demonstrate the advantages of AdaRoute.
>
>
> >SNW2&ORW1: Stage-wise vs. Global Expert Center
>
> A2: Both Tab.6 (AP^b: 46.3 to 47.3 as the sharing scope grows from 3 to 18 layers) and Sec.C.2 (Eqs.1 to 3) confirm that a larger sharing scope strengthens cross-layer interaction. The center is stage-wise because hierarchical architectures use different feature dimensions per stage (128, 256, 512, 1024 in Swin-B). A global center requires additional dimension alignment operations, adding complexity that undermines PEFT simplicity. How to make dimension alignment efficient in a global center warrants a promising future direction. The current design maximizes interaction within each stage.
>
>
> >SNW3: Spatial Dependency and Performance Attribution
>
> A3: The reviewer identifies three factors: spatial dependency (SD), representational diversity (DR), and dynamic routing (RT). We clarify that **these are not independent but emergent properties of one mechanism**: dynamic routing over a shared expert center. The router aggregates experts into dynamic weights ($W_l = \sum_{m=1}^{M} g_l^m \mathcal{E}_m$, Eq.2), simultaneously resulting in: (1) input-dependent weights yielding a larger ERF (Fig.1(c), row 1), and (2) cross-layer gradient aggregation (Eq.3, Sec.C.2) promoting feature diversity, yielding lower CKA similarity (Fig.1(c), row 2).
>
> To disentangle the contributions, we construct AdaRoute* by replacing dynamic multi-scale convolutions with Mona's standard multi-scale convolutions. This variant no longer possesses dynamic spatial modeling capability and retains only dynamic channel modeling, whereas Mona lacks both capabilities. We also report Boundary IoU (B-IoU) to evaluate boundary precision. Experiments are conducted on domain adaptation segmentation to better reflect the robustness of different models:
>
> | Method | mIoU | B-IoU |
> |---|---|---|
> | AdaRoute (dynamic channel&spatial modeling) | 63.6 | 55.1 |
> | AdaRoute* (w/o dynamic spatial) | 62.5 | 54.1 |
> | Mona (w/o dynamic spatial&channel) | 61.9 | 53.5 |
>
> Dynamic channel modeling (Mona->AdaRoute*) improves by 0.6/0.6 mIoU/B-IoU. Adding dynamic spatial mixing (AdaRoute*->AdaRoute) contributes 1.1/1.0 mIoU/B-IoU. This suggests that better spatial modeling not only improves overall recognition but also enhances boundary perception.
>
>
> >PW1-PW3: ERF&CKA Details
>
> A4: Both ERF and CKA in Fig.1(c) are averaged over 100 randomly sampled COCO2017 validation images based on the complete model (using more images yields negligible changes).
>
> >PW4, PW5, PW8: Figure Improvements
>
> A6: We will carefully improve figure placement, optimize the digits, and add visual anchors across figures for readability.
>
> >PW6: VPT Params
>
> A7: As noted in Sec.4 (p.5), VPT appends trainable tokens along the spatial dimension, directly increasing the quadratic complexity of self-attention. On hierarchical models (Swin), scaling VPT to adapter-comparable parameters causes OOM on our side, so we retain VPT's original configuration. In Tab.1-3, #P only counts trainable parameters of the backbone since the decoder is identical across all methods, yielding 0.1-0.2M for VPT. In Tab.7, the backbone is ViT-L with the original VPT setup, but #P includes the segmentation decoder, which accounts for 13.6M. We sincerely apologize for not specifically mentioning this, and we will clarify it in the revised version.
>
> >PW7: AdaRoute Params
>
> A8: We report **all trainable parameters** of the AdaRoute-equipped backbone: expert matrices in all stages and router networks. No trainable parameters are excluded.
>
> >Limitations
>
> A9: We thank the reviewer for noting potential adversarial attacks to the routing mechanism. We will add this discussion to the limitation section.
>
> We sincerely hope our responses have thoroughly addressed your concerns and would deeply appreciate it if you could increase your score in light of our revisions.

---

> > ### Author Rebuttal · Reviewer_MaEV · 2026-04-01
> >
> > Thanks for the detailed response. Most of the concerns have been addressed especially __SN_W3__.

---

> > > ### Author Response · Authors · 2026-04-05
> > >
> > > We sincerely appreciate the reviewer's decision to raise the score (``4-->5``). We will carefully incorporate the additional results and discussions into the revised manuscript to further strengthen the paper. Once again, many thanks for your meticulous review and insightful comments.

---

### Official Review · Reviewer_24T9 · 2026-03-02

**Soundness:** 3
**Presentation:** 3
**Significance:** 3
**Originality:** 2
**Overall Recommendation:** 4
**Confidence:** 4

**Summary:**

This paper proposes AdaRoute, a parameter-efficient fine-tuning (PEFT) framework for adapting pretrained vision models to challenging dense prediction tasks, including semantic segmentation, object detection, and panoptic segmentation. AdaRoute introduces a mixture-of-experts (MoE) design by constructing a shared expert center composed of trainable parameter matrices and employing lightweight, input-dependent routers to dynamically aggregate these experts. This dynamic parameter routing enables input-adaptive low-rank transformations within each layer’s adapter, while cross-layer sharing of the expert center encourages diverse and non-redundant feature learning. Extensive experiments across multiple vision tasks show that AdaRoute matches or surpasses strong PEFT baselines. Ablation studies and visualization analyses further elucidate the impact of key design components.

**Compliance With Llm Reviewing Policy:**

Affirmed.

**Final Justification:**

The authors’ responses have partially addressed my concerns. In particular, some issues—such as the lack of more comprehensive experimental validation—remain to be further explored in future work. Therefore, I decide to maintain my original score.

**Key Questions For Authors:**

1 How does AdaRoute compare to mixture-of-expert PEFT architectures like MoA and MoLEx on complex vision tasks, particularly in dense prediction scenarios? Is there any reason to believe AdaRoute’s cross-layer expert sharing confers unique benefits?

2 Can you provide results or discussion for AdaRoute on larger/foundation models (e.g., DINOv3, ViT-Giant) and elaborate on expected scalability? What are the main bottlenecks for extending to these scales?

3 What are the consequences for inference-time costs in deployment? Is there a trade-off between dynamic routing flexibility and speed in real-world applications?

**Limitations:**

Please refer to the above Strengths and Weaknesses section for comments regarding the limitations of the work.

**Strengths And Weaknesses:**

Strength:

1 The paper proposes a novel MoE-style fine-tuning framework with input-adaptive routing, introducing a learnable and dynamic parameter adaptation mechanism. The method achieves competitive state-of-the-art performance across multiple dense prediction tasks.

2 The manuscript is clearly written and easy to follow. The overall architecture is conceptually straightforward and avoids unnecessary complexity, which enhances reproducibility.

3 The input-adaptive dynamic routing mechanism is particularly interesting. Its integration with parameter-efficient fine-tuning is well motivated, and the effectiveness of the individual components is empirically validated through comprehensive ablation studies.

Weakness:

1 Although empirically effective, the core idea—introducing additional shared learnable parameters with input-dependent routing—has been extensively explored in prior work. Related paradigms include dynamic parameterization and conditional adapters, such as **Dynamic Filter Networks**, **HydraLoRA: An Asymmetric LoRA Architecture for Efficient Fine-Tuning**, **Mixture of Cluster-conditional LoRA Experts for Vision-Language Instruction Tuning**, and **CondConv: Conditionally Parameterized Convolutions for Efficient Inference**. The manuscript does not sufficiently articulate the conceptual novelty or fundamental differences of AdaRoute relative to these existing approaches.

2 While experiments are conducted on several segmentation and detection benchmarks with multiple visual backbones, the robustness and generalization capability of the method remain unclear. For example, it would be valuable to evaluate performance under more challenging settings such as domain adaptation or domain generalization to assess whether the adaptive routing mechanism provides robustness benefits beyond standard supervised fine-tuning.

3 The adaptability of the method to a broader range of large-scale vision foundation models requires further validation. It remains unclear whether AdaRoute can be seamlessly integrated into widely used pretrained models such as DINO, Segment Anything, or CLIP while retaining its effectiveness. Demonstrating compatibility with such models would significantly strengthen the practical relevance and impact of the work.

---

> ### Author Rebuttal · Authors · 2026-03-27
>
> >W1&Q1: Comparison with Related Methods
>
> A1: Thank you for this thoughtful suggestion. We would like to clarify that our method has fundamental distinctions from the ones mentioned by the reviewer:
>
> ``DFN`` and ``CondConv`` require a separate and large weight generation network at every layer to generate complete filter weights. These per-layer weight generation networks are not shared across layers, resulting in substantial parameter overhead unsuitable for PEFT settings, which emphasize parameter efficiency and architectural simplicity. In contrast, AdaRoute maintains a shared expert center in a network stage and lets each layer selectively aggregate experts through a lightweight routing mechanism. This avoids appending complex modules with a large number of trainable parameters at every layer, thereby maintaining the efficiency required by PEFT.
>
> ``HydraLoRA`` and ``MoClE`` dynamically combine the output features of multiple expert branches (**output features serve as experts**), while the weight matrices within each branch remain input-independent. AdaRoute instead dynamically composes weight matrices from shared expert centers (**parameters serve as experts**), producing input-dependent transformations that vary per input.
>
> More importantly, **none of the above methods shares experts across different network layers**. Our shared expert center creates implicit cross-layer interaction through multi-layer gradient aggregation, which is the key mechanism for reducing feature redundancy as evidenced by the CKA analysis in Fig.1(c). This also differentiates AdaRoute from other MoE-based PEFT methods (more details are given in Sec. C.2).
>
> In summary, the novelty of AdaRoute lies not in simply applying dynamic parameterization to PEFT, but in a unified design (shared expert center + dynamic routing) that simultaneously enables both input-dependent adaptation and cross-layer interaction. This differs from previous works fundamentally.
>
> As you suggested, we further adapt MoA and MoLEx to semantic segmentation (Swin-B + UperNet, ADE20K): our AdaRoute surpasses both methods by a clear margin.
>
> | Method | #P (M) | mIoU |
> |---|---|---|
> | MoA | 5.6 | 48.7 |
> | MoLEx | 5.5 | 48.9 |
> | AdaRoute | 5.2 | **50.3** |
>
> >W2: Robustness Under Domain Shift
>
> A2: We appreciate this insightful comment. We conduct domain adaptation experiments on ``Cityscapes->Foggy Cityscapes`` using AdaptSegNet+Swin-B:
>
> | Method | #P (M) | mIoU |
> |---|---|---|
> | Full-tuning | 86.8 | 64.1 |
> | Mona | 5.2 | 61.9 |
> | AdaRoute | 5.2 | 63.6 |
>
> Under severe domain shifts, AdaRoute narrows the gap with full-tuning to 0.5 mIoU while Mona trails by 2.2 mIoU. The +1.7 gain over Mona suggests that our method is more robust, i.e., when encountering out-of-distribution inputs, the router can adjust expert aggregation accordingly, whereas input-independent adapters lack this flexibility. We appreciate the helpful suggestion, which further validates the robustness of our method in handling domain shifts.
>
> >W3&Q2: Scalability to Larger Models
>
> A3: We appreciate this thoughtful comment. We conduct semantic segmentation on ADE20K using OpenCLIP-ConvNeXt-XXL-LAION2B:
>
> | Method | #P (%) | mIoU |
> |---|---|---|
> | Full-tuning | 100% | 52.4 |
> | Mona | 2.2% | 54.0 |
> | AdaRoute | 2.2% | **54.8** |
>
> First, full-tuning suffers from overfitting on the XXL model, falling behind both PEFT methods. Second, AdaRoute maintains a notable advantage over Mona, indicating robust scalability. Meanwhile, our paper has already validated architectural generality across Swin, ConvNeXt, ViT, and VMamba. The above result further extends this validation to a larger scale. Due to limited resources, we are unable to conduct experiments on DINOv3-ViT-G and SAM-ViT-H. That is, computational resources are the primary bottleneck, and extending PEFT methods to this scale remains a community-wide challenge. We will further explore scaling up to larger foundation models with better efficiency and lower computational cost in the future.
>
>
> >Q3: Deployment Latency
>
> A4: Thank you for this insightful suggestion. In fact, we have provided an efficiency analysis in Tab.15 (p.16). Here, we further present latency comparisons among different ONNX-deployed models based on Swin-L, using an input of 1x3x1280x800 and averaging over 100 iterations:
>
> | Method | Latency (ms) | AP^b |
> |---|---|---|
> | RepAdapter | 114 | 46.9 |
> | SNELL | 112 | 41.3 |
> | Mona | 122 | 48.1 |
> | AdaRoute‡ | 115 | 47.7 |
> | AdaRoute | 122 | 48.6 |
>
> AdaRoute improves Mona by 0.5 AP^b at the same speed. Meanwhile, AdaRoute‡ (w/o multi-scale convs) runs at 115 ms with 47.7 AP^b, improving RepAdapter by 0.8 AP^b with comparable efficiency. This suggests that our method offers a favorable tradeoff between deployment efficiency and performance.
>
> We sincerely hope our responses have thoroughly addressed your concerns and would deeply appreciate it if you could increase your score in light of our revisions.

---

> > ### Author Rebuttal · Reviewer_24T9 · 2026-04-01
> >
> > The authors’ response has addressed my concerns to a certain extent. While the experimental evaluation could be further strengthened in future work, I will maintain my original score. I also thank the authors for their clarifications and additional experimental details.

---

> > > ### Author Response · Authors · 2026-04-05
> > >
> > > We sincerely appreciate the reviewer’s positive assessment and recommendation. We will carefully integrate the additional evaluations and discussions into the revised manuscript to further strengthen the paper. Once again, many thanks for your meticulous review and insightful comments.

---

### Official Review · Reviewer_ykuU · 2026-03-08

**Soundness:** 3
**Presentation:** 3
**Significance:** 3
**Originality:** 3
**Overall Recommendation:** 3
**Confidence:** 4

**Summary:**

The authors propose a novel visual PEFT method that optimizes the parameter fine-tuning process in visual adapters. Experiments on multiple datasets demonstrate that the proposed method can effectively improve visual adapters such as Mona.

**Compliance With Llm Reviewing Policy:**

Affirmed.

**Final Justification:**

As mentioned, I will keep my score, while actually neutral about accepting or not

**Key Questions For Authors:**

-

**Limitations:**

-

**Strengths And Weaknesses:**

Strengths:
1. The figures and tables are well-prepared.
2. The experiments are comprehensive.
3. The topic is meaningful and valuable.

Weakness:
1. The authors primarily demonstrate the advantages of the proposed method through experimental results, which is common in vision conferences such as CVPR. Given that this is ICML, I strongly prefer the authors to provide solid theoretical analysis to explain why the proposed method outperforms Mona.
2. I have concerns about some of the experimental results. Both the authors and Mona appear to conduct experiments with a batch size of 16. Mona reports 53.4 on COCO/Swin-B in its original paper, but Table 2 here shows 46.6. The authors are requested to provide a necessary explanation or additional experimental results.
3. Code is not provided in the supplementary materials, which affects my evaluation of the paper.
4. Since this work is an incremental improvement over Mona, it is recommended to align with the datasets used in Mona (e.g., DOTA).

---

> ### Author Rebuttal · Authors · 2026-03-27
>
> >W1: Theoretical Analysis
>
> A1: We appreciate this thoughtful suggestion. In fact, we have provided a theoretical analysis in Sec. C.2 (p.18), which we briefly summarize here for your convenience. For more details, please refer to Sec. C.2.
>
> In conventional PEFT methods (e.g., AdaptFormer and Mona), each layer $l$ has an independent adapter $W_l$, yielding a gradient (Eq.1):
>
> $$\frac{\partial \mathcal{L}}{\partial W\_l} = \frac{\partial \mathcal{L}}{\partial h\_l} \cdot \frac{\partial h_l}{\partial W\_l}$$
>
> This gradient does not explicitly account for local gradients from other layers. Although a gradient update at an earlier layer can directly alter its own output representations, its effect on deeper layers must propagate sequentially through the network forward pass.
>
> In AdaRoute, each effective weight $W_l$ is constructed via a shared expert center $E = \{\mathcal{E}_1, \dots, \mathcal{E}_M\}$ (Eq. 2):
>
> $$W_l = \sum_{m=1}^{M} g_l^m \mathcal{E}_m$$
>
> The gradient of expert $\mathcal{E}_m$ then explicitly aggregates layer-wise contributions across all $L$ layers (Eq. 3):
>
> $$\frac{\partial \mathcal{L}}{\partial \mathcal{E}\_m} = \sum_{l=1}^{L} \frac{\partial \mathcal{L}}{\partial h\_l} \cdot \frac{\partial h_l}{\partial W\_l} \cdot \frac{\partial W\_l}{\partial \mathcal{E}\_m}$$
>
> This is the key theoretical distinction: updating $\mathcal{E}_m$ holistically integrates gradient information flowing through all network layers sharing the expert. Once $\mathcal{E}_m$ is updated, the weight matrices of all layers sharing $\mathcal{E}_m$ are immediately affected, thereby enabling more direct modulation of feature representations in each layer. This establishes an **implicit cross-layer interaction** without relying on memory-costly explicit feature connections like DenseNet. This mechanism is also a key difference from other PEFT methods, such as Mona (Sec. C.3), where adapter weights remain layer-isolated.
>
> *Although this theoretical framework seems straightforward, it reflects the fundamental reason why AdaRoute outperforms other PEFT methods: enhancing feature diversity compared to other methods (intuitively shown in Fig.1)*. Also, this analysis is consistent with our empirical findings: Tab.6 shows that broader expert sharing scope consistently improves performance (AP^b: 46.3->47.3 as scope grows from 3 to 18 layers), and Tab.13 confirms that sparse expert activations, which weaken cross-layer coupling, cause performance degradation.
>
> >W2: Experimental Results
>
> A2: We appreciate your meticulous review. The discrepancy arises from two factors:
>
> **Different detectors.** Mona's original paper uses more powerful but heavier **Cascade Mask R-CNN 3x schedule (36 epochs)**, whereas our Tab.2 uses **Mask R-CNN 1x schedule (12 epochs)**. We chose Mask R-CNN because PEFT modules operate in the backbone, a simpler detector better reflects backbone representation quality. In addition, the 1x schedule requires fewer computational resources, which better aligns with the parameter-efficient setting.
>
> **Hidden rank alignment for fair comparison.** As mentioned in Sec. 4, we slightly increase the hidden rank of Mona to align all PEFT methods with comparable parameters for fair comparisons. This explains why Mona's reproduction under the same segmentation framework (UperNet) is higher than the original reported numbers (51.4->51.6 mIoU).
>
> To directly address this concern, we provide Cascade Mask R-CNN results under comparable parameters:
>
> | Swin-B Cas. (3x) | #P (M) | AP^b | AP^m |
> |---|---|---|---|
> | Mona | 5.2 | 53.8 | 46.2 |
> | AdaRoute | 5.2 | 54.9 (+1.1) | 46.8 (+0.6) |
>
> Notably, under the more robust detector with a longer training schedule, AdaRoute yields even clearer improvements, suggesting that stronger detectors better leverage richer backbone representations empowered by AdaRoute.
>
> >W3: Source Code
>
> A3: We deeply acknowledge that code is critical for reproducibility. As the rebuttal policy banning code links, we commit to releasing our code publicly upon acceptance. As rebuttals will be visible with accepted manuscripts, this statement forms our binding commitment.
>
> >W4: Results on DOTA
>
> A4: We appreciate this insightful comment. At first, we would like to clarify that AdaRoute introduces two structurally novel properties absent in Mona and prior visual PEFT: (1) implicit cross-layer interaction via shared expert centers with multi-layer gradient aggregation (Eq. 3); and (2) input-dependent low-rank adaptation where effective weights are dynamically composed per input (Sec. 3.2).
>
> | Swin-B+Oriented R-CNN | #P (M) | AP |
> |---|---|---|
> | Mona | 5.2 | 78.7 |
> | AdaRoute | 5.2 | 79.4 (+0.7) |
>
> From the above results, AdaRoute consistently surpasses Mona on DOTA1.0, further confirming the robustness of our method.
>
> We sincerely hope our responses have thoroughly addressed your concerns and would deeply appreciate it if you could increase your score in light of our revisions.

---

> > ### Author Rebuttal · Reviewer_ykuU · 2026-04-03
> >
> > Thank you for your response. Issues 2 and 4 have been addressed, but I still think the theoretical depth of this paper is insufficient for ICML, and I will retain my original score. In addition, the code can be submitted as supplementary material at the initial stage or via an anonymous GitHub link.

---

> > > ### Author Response · Authors · 2026-04-05
> > >
> > > We greatly appreciate the reviewer for confirming that issues 2 and 4 have been resolved and for the constructive engagement throughout this process. We respectfully address the two remaining points below.
> > >
> > >
> > > First, we agree that theoretical analysis can always be further deepened. However, we believe that the primary goal of theoretical analysis is to provide clear and accessible mathematical insight that enables the readers to understand the working mechanism behind a method and to build upon it. In this work, we have provided a clean theoretical analysis explicitly identifying the core working mechanism of AdaRoute, which is empirically validated through extensive ablation studies and visualizations (Fig.1, Tab.6, and Tab.13).
> > >
> > >
> > > Beyond theoretical analysis, this paper has presented comprehensive evaluations, covering semantic segmentation, object detection, instance segmentation, panoptic segmentation, image classification, pose estimation, and remote sensing segmentation. These experiments are conducted on a diverse set of widely used backbones, with consistent performance gains observed across all settings. Notably, the other three reviewers have provided detailed technical evaluations of this work and acknowledged our rebuttal responses positively. On method, Reviewer 24T9 noted that ``"the overall architecture is conceptually straightforward and avoids unnecessary complexity, which enhances reproducibility"``, and Reviewer 7skm commented that ``"the method is sensible and logical"``. On analysis, Reviewer MaEV recognized that "``this work employs Centered Kernel Alignment (CKA) analysis to systematically assess feature redundancy"`` and ``"It is good to use effective receptive field (ERF) to measure representation deficiency compared to existing works"``.  On experiments, Reviewer 7skm stated that ``"the experiments are extremely thorough and convincing"`` and that ``"the paper provides comprehensive metrics, including trainable parameters, GPU memory, and training throughput, rather than just reporting accuracy"``.
> > >
> > >
> > > Regarding code availability, we would like to clarify that, according to the ICML'26 guidelines, code submission at the initial stage is **not a requirement**. The rebuttal guidelines do not permit sharing code via anonymous links during this phase. In addition, we have provided clear implementation details to ensure reproducibility. Meanwhile, we have already made a formal commitment in our rebuttal to release the source code.
> > >
> > >
> > > Overall, we fully respect your final assessment and sincerely appreciate the valuable feedback.

---

### Decision · Program_Chairs · 2026-04-30

**Decision:**

Accept (regular)

**Comment:**

The paper addresses the task of dense prediction and proposes AdaRoute, a new adapter-style method featuring a simple mixture-of-experts architecture. The proposed method achieves favorable results on key vision tasks. The rebuttal was engaging and clarified a number of reviewers’ concerns, with some of them raising their recommendation after the rebuttal. In the end, three reviewers are positive about accepting the paper, while the fourth one does not oppose it.